# The Hippo pathway transcription factors YAP and TAZ play HPV-type dependent roles in cervical cancer

Molly R. Patterson [1,2] ✉, Joseph A. Cogan [1,2], Rosa Cassidy[1,2], Daisy A. Theobald[1,2], Miao Wang[1,2], James A. Scarth [3], Chinedu A. Anene [3,4], Adrian Whitehouse [1,2], Ethan L. Morgan [5] & Andrew Macdonald [1,2] ✉

Human papillomaviruses (HPVs) cause most cervical cancers and an increasing number of anogenital and oral carcinomas, with most cases caused by HPV16 or HPV18. HPV hijacks host signalling pathways to promote carcinogenesis. Understanding these interactions could permit identification of much-needed therapeutics for HPV-driven malignancies. The Hippo signalling pathway is important in HPV+ cancers, with the downstream effector YAP playing a pro-oncogenic role. In contrast, the significance of its paralogue TAZ remains largely uncharacterised in these cancers. We demonstrate that TAZ is dysregulated in a HPV-type dependent manner by a distinct mechanism to that of YAP and controls proliferation via alternative cellular targets. Analysis of cervical cancer cell lines and patient biopsies revealed that TAZ expression was only significantly increased in HPV18+ and HPV18-like cells and TAZ knockdown reduced proliferation, migration and invasion only in HPV18+ cells. RNA-sequencing of HPV18+ cervical cells revealed that YAP and TAZ have distinct targets, suggesting they promote carcinogenesis by different mechanisms. Thus, in HPV18+ cancers, YAP and TAZ play non-redundant roles. This analysis identified *TOGARAM2* as a previously uncharacterised TAZ target and demonstrates its role as a key effector of TAZ-mediated proliferation, migration and invasion in HPV18+ cancers.

Cervical cancer remains a global public health problem, with the fourth most common incidence and mortality rates in women worldwide, accounting for ~604,000 new cases and ~340,000 deaths in 2020[1]. Persistent infection with high-risk human papillomavirus (HR-HPV) type 16 accounts for most cervical cancers (~50%), with HPV18 being the second most common type, accounting for ~12% of squamous cell carcinomas as well as 37% of adenocarcinomas of the cervix[2].

Although our understanding of HPV-mediated transformation has increased in recent years, there are currently no specific therapeutics targeting HPV or HPV-driven cancer. Thus, more research is required to identify ways to treat these cancers. Identifying how cell signalling pathways are dysregulated in HPV-driven malignancies may provide such therapeutic options[2–5].

The Hippo signalling pathway is a regulator of organ size and development, controlling the transcription of genes involved in cell growth and proliferation[6]. Although many stimuli can activate the pathway, the core kinase cascade consists of STK4/3 and LATS1/2 (together with SAV1 and MOB1) and culminates with phosphorylation

[1]School of Molecular and Cellular Biology, Faculty of Biological Sciences, University of Leeds, Leeds LS2 9JT, UK. [2]Astbury Centre for Structural Molecular Biology, University of Leeds, Leeds, UK. [3]Barts Cancer Institute, Queen Mary University of London, London, UK. [4]Centre for Biomedical Science Research, Leeds Beckett University, Leeds, UK. [5]School of Life Sciences, University of Sussex, Brighton, UK. ✉e-mail: molly.patterson@pennmedicine.upenn.edu; a.macdonald@leeds.ac.uk

and subsequent inactivation and degradation of the transcription factors YAP and TAZ[7,8]. Recent studies have highlighted the importance of this pathway evidenced by the crucial oncogenic functions of YAP[9]. Both HR-HPV E6 and E7 promote YAP stabilisation and activation, emphasising its importance in cervical cancer[10–13]. The beta papillomavirus HPV8 E6 can also induce YAP-dependent transcriptional activity, potentially through reducing activity of its negative regulator LATS1/2[14]. Importantly, Hippo pathway dysregulation is a major driving feature of cervical carcinogenesis, as hyperactivation of YAP is sufficient to drive the development of cervical squamous cell carcinoma (CSCC) in the absence of HPV[15]. In contrast, much less is known about the role of the YAP paralogue TAZ in cancers, as until recently it had been broadly assumed to play an interchangeable role to YAP. However, emerging evidence supports YAP-independent functions for TAZ[16].

A small number of previous studies have analysed the role of TAZ in cervical cancer; however, their findings are controversial. One study identified that TAZ expression is decreased in cervical cancer compared with normal tissue[17]. In contrast, others suggest that TAZ is oncogenic and regulates proliferation, invasion and apoptosis whilst also upregulating PD-L1, a prominent regulator of T-cell evasion[18]. Thus, further research is required to better understand the role of TAZ within HPV+ cervical cancer.

In this study, we reveal distinct mechanisms of YAP and TAZ regulation by HPV. While YAP protein expression was increased in both HPV16+ and HPV18+ cervical cancer, TAZ protein expression was only upregulated in HPV18+ cervical cancer. Furthermore, we established that increased TAZ protein expression was driven by increased *WWTR1* (the gene encoding TAZ) promoter activity likely through an ERK1/2-SP1 signalling pathway in an HPV18 E7-dependent manner. Reducing TAZ activity, either by pharmacological inhibition or shRNA-mediated knockdown (KD), significantly impeded proliferation, migration and invasion in HPV18 +, but not HPV16 +, cervical cancer cells. Importantly, we showed that YAP overexpression did not rescue the proliferation defect observed in TAZ-depleted cells and that YAP and TAZ have distinct transcriptional target profiles, highlighting their non-redundant roles in HPV18-driven cancers. We identify the poorly understood gene *TOGARAM2* as a TAZ target that plays a critical role in mediating the oncogenic functions of TAZ in HPV18+ cervical cancer cells. Taken together, we show that TAZ functions as an oncogene in a HPV type-specific manner and demonstrate that YAP and TAZ play non-redundant roles in cervical cancer. Furthermore, we provide evidence that *TOGARAM2* is an oncogenic TAZ-specific gene, essential for TAZ-dependent proliferation, migration and invasion of tumour cells.

## Results

### TAZ is upregulated in HPV18+ and HPV18-like+ cervical cancers
Given the lack of an agreed role for TAZ in cervical carcinogenesis, we analysed expression of the gene encoding TAZ (*WWTR1*) in The Cancer Genome Atlas (TCGA) dataset. *WWTR1* was amplified in around 13% of cervical cancer patients (Fig. 1A). Further analysis demonstrated positive correlation of *WWTR1* copy number with mRNA expression ($R = 0.256$) and *WWTR1* mRNA expression also positively correlated with TAZ protein expression ($R = 0.181$) (Fig. 1B, C). To validate these findings, we investigated YAP and TAZ protein expression in a panel of cervical cancer cell lines. Previous studies have shown that YAP expression is elevated in both HPV16+ and HPV18+ cervical cancer cell lines[10]. Interestingly, when compared to immortalised but non-transformed keratinocytes (HaCaT cells) and HPV negative (-) cervical cancer cells (C33A), we only observed a significant increase in TAZ protein levels in HPV18+ (HeLa and C4-1) and HPV45+ (HPV18-related alpha-7 species) (MS751) cell lines (Fig. 1D). YAP levels are increased in HPV-driven cancers due to a proposed increase in protein stability[11,12], which we confirmed with a cycloheximide (CHX) chase assay in HeLa cells. While YAP protein levels increased compared to GAPDH, in the presence of CHX, TAZ levels rapidly reduced with a half-life of

approximately 2 h, suggesting that YAP and TAZ might be regulated differently in HPV18+ cells (Fig S1A–B). Further analysis revealed *WWTR1* mRNA levels were increased in HPV18 +/HPV45+ but not HPV16+ cervical cancer cell lines. In contrast, *YAP1* mRNA levels did not differ significantly between any of the cell lines analysed (Fig. 1E). We extended our study to patient liquid-based cervical cytology samples collected at distinct stages of cervical intraepithelial neoplasia (CIN) from a cohort of HPV16+ or HPV18+ patients. CIN lesions are precursors to cervical carcinomas and ideal for studying the progression towards cervical cancer. We found *WWTR1* mRNA to be significantly increased in the CIN2/3 samples taken from HPV18+ patients, but not HPV16+ patients (Fig. 1F). Finally, we investigated the clinical relevance of *WWTR1* expression in cervical cancer patients using the TCGA cohort. High *WWTR1* expression correlated with reduced progression-free survival specifically in HPV18+ patients, when compared with non-HPV18+ cervical cancer patients (those harbouring other HR-HPV types or HPV-) (Fig. 1G, H).

### HPV18 E7 increases *WWTR1* transcription
To gain an understanding of how *WWTR1*/TAZ expression is increased in HPV18+ cervical cancer cells we tested the contribution of the HPV oncoproteins. Expression of HPV18 E6/E7 was knocked down in HeLa cells using siRNA, which resulted in a significant decrease in TAZ expression (Fig. 2A, B). Interestingly, despite the reduction in TAZ levels we did not observe a similar decrease in YAP protein expression in these experiments (Fig. 2B). Next, we assessed whether this was due to a reduction in the activity of the *WWTR1* promoter using a luciferase reporter assay. This revealed that the reduced HPV18 E6/E7 expression decreased promoter activity (Fig. 2C). Conversely, overexpression of HPV18 E7 in C33A cells led to a 3-fold increase in *WWTR1* mRNA expression coupled with increased promoter activity and TAZ protein expression, while overexpression of HPV18 E6 led to no significant change (Fig. 2D–F). Similar results were obtained in HaCaT cells (Fig S2 A–C). As we had observed increased TAZ expression in HPV18+ but not HPV16+ cervical cancer cells, we investigated if only HPV18 E7 was able to induce TAZ expression. In contrast to HPV18 E7, overexpressed HPV16 E7 did not significantly increase *WWTR1* promoter activity, mRNA expression or TAZ protein expression (Fig. 2G–I). Further, HPV16 E6/E7 knockdown did not reduce WWTR1/TAZ mRNA or protein levels from HPV16+ cell lines (Fig S2 D, E). Taken together, these data reveal that HPV18 E7 increases TAZ levels through increased promoter activity.

### HPV18 E7-mediated TAZ expression requires the SP1 transcription factor
Analysis of the *WWTR1* promoter revealed the presence of two potential SP1 binding sites (Fig. 3A). Since HPV E7 can regulate SP1[19,20], we investigated the impact of perturbing SP1 transactivation activity on TAZ protein expression. Treatment with the SP1 inhibitor mithramycin A (Mith A)[21] reduced both *WWTR1* promoter activity and *WWTR1* mRNA expression, whilst having minimal effect on *YAP1* mRNA expression (Fig. 3B, C). The specific effect of Mith A on *WWTR1* mRNA expression was also observed in the HPV18 + SW756 and related HPV45 + MS751 cancer cell lines (Fig S3A, B), arguing against a cell line-specific effect. Crucially, Mith A treatment prevented the increase in *WWTR1* promoter activity, mRNA accumulation and TAZ protein expression observed following transient HPV18 E7 overexpression (Fig. 3D, F). Expression of a dominant negative SP1 truncation mutant lacking the transactivation domain (trunc-SP1)[22] caused a similar decrease in *WWTR1* promoter activity, mRNA abundance and TAZ protein expression (Fig S3C, E). Furthermore, deletion of either of the two SP1 binding motifs in the *WWTR1* promoter significantly reduced both endogenous and HPV18 E7-induced promoter activity, providing further support for SP1 regulating *WWTR1*/TAZ expression in HPV18+ cells (Fig. 3G, H).

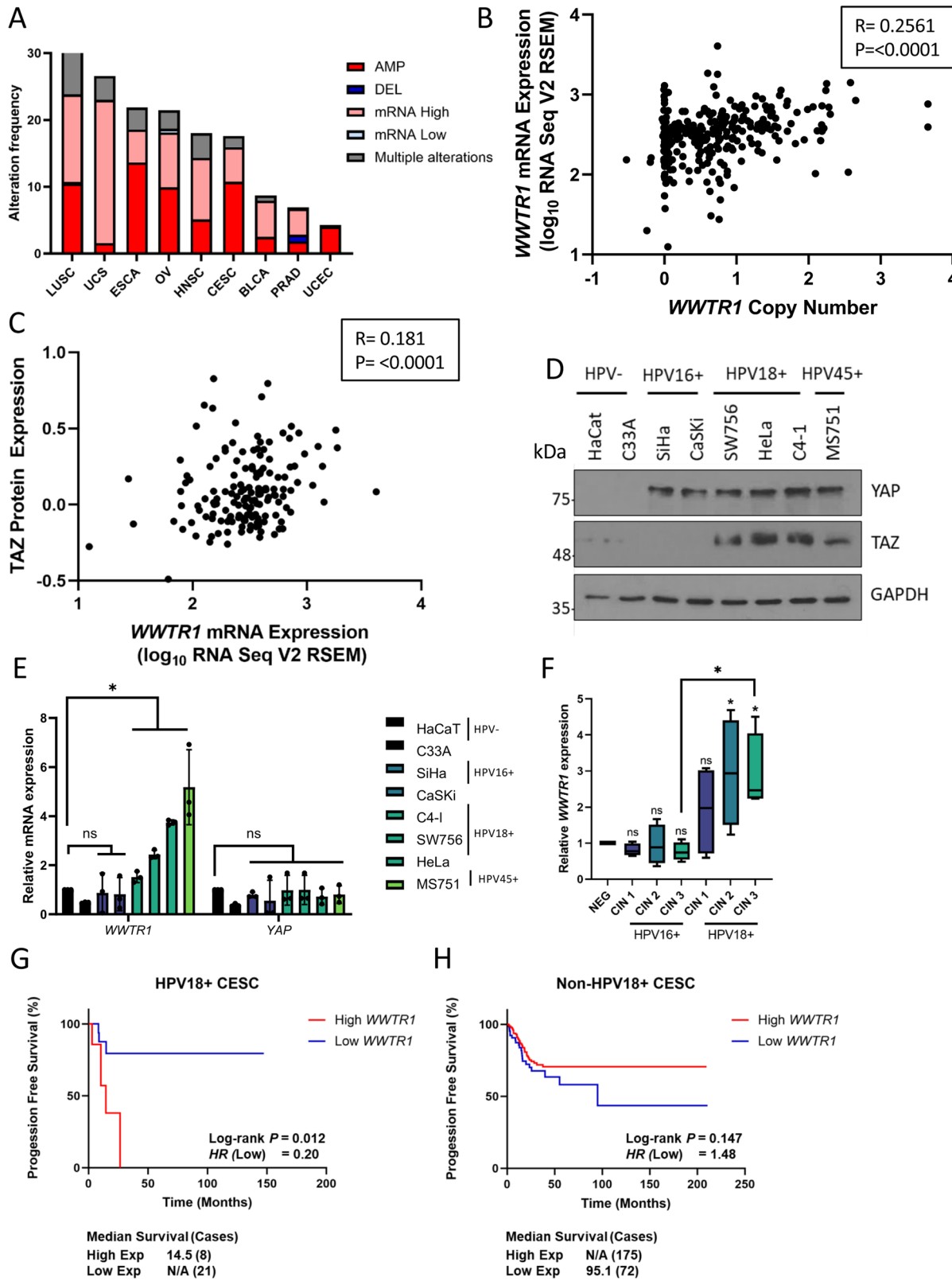

SP1 is positively regulated by ERK1/2 MAPK-dependent phosphorylation of T453 and T739[23], and mutation of these residues to aspartic acid results in a constitutively active form of SP1 (SP1 2TD). When overexpressed in HeLa cells depleted of E6/E7 expression, this dominant active SP1 rescued *WWTR1* promoter activity and mRNA expression (Fig. 3I, J and S3F). Conversely, pharmacological blockade of ERK1/2 activity using the MEK1/2 inhibitor U0126 reduced TAZ

levels in E7-expressing C33A cells (Fig. 3K, M). Interestingly, whilst ERK1/2 blockade had a significant negative impact on TAZ protein levels, we did not observe any significant reduction in YAP protein levels under these conditions. Finally, we assessed if SP1 regulated TAZ expression directly by binding to the *WWTR1* promoter via the two putative SP1 binding sites. We performed ChIP-qPCR utilising primers covering the two predicted SP1 binding sequences

**Fig. 1 | TAZ is upregulated in HPV18+ cervical cancer. A** Genomic alterations of *WWTR1* across human cancers determined by cBioportal analysis of TCGA data (LUSC Lung squamous cell carcinoma (*n* = 487), USC Uterine serous carcinoma (*n* = 56), ESCA Eosophageal carcinoma (*n* = 186), OV Ovarian serous cystadenocarcinoma (*n* = 579), HNSC Head and Neck squamous cell carcinoma (*n* = 517), CESC Cervical squamous cell carcinoma (*n* = 295), BLCA Bladder Urothelial carcinoma (*n* = 408), PRAD Prostate adenocarcinoma (*n* = 491), UCEC Uterine corpus endometrial carcinoma (*n* = 487). **B** Scatter dot plot analysis of *WWTR1* mRNA expression against *WWTR1* copy number alterations in cervical cancer determined from TCGA data (*n* = 278). The correlation coefficient (r) was calculated using Pearson correlation analysis. **C** Scatter dot plot analysis of *WWTR1* mRNA expression against TAZ protein expression in cervical cancer determined from TCGA data (*n* = 152). The correlation coefficient (r) was calculated using Pearson correlation analysis. **D** Representative western blot of lysate from HPV-, HPV16+ and HPV18+ cell lines. Lysates were probed for YAP, TAZ and the loading control GAPDH (*n* = 3). **E** RT-

qPCR analysis of *WWTR1* or *YAP1* expression in HPV-, HPV16+ or HPV18+ cell lines (*n* = 3). *U6* transcript levels were used as a loading control. **F** RT-qPCR analysis of *WWTR1* expression in negative, HPV16 or HPV18+ patient cervix liquid cytology samples from different CIN grades (*n* = 4 from each grade). *U6* was used as a loading control. Maxima and minima whiskers represent the highest and lowest values, respectively. Bounds of boxes are the 75th and 25th percentiles and the centre represents the median. **G** Kaplan−Meier curves showing progression free survival in HPV18+ cervical cancer stratified by high or low *WWTR1* expression. Survival was compared using the log-rank test. **H** Kaplan−Meier curves showing progression free survival in non-HPV18+ cervical cancer stratified by high or low *WWTR1* expression. Survival was compared using the log-rank test. Error bars represent the mean +/- standard deviation of a minimum of three biological repeats when relevant. *\*P* < 0.05, *\*\*P* < 0.01, *\*\*\*P* < 0.005 (two-tailed, unpaired Student's *t*-test). Source data are provided as a Source Data file.

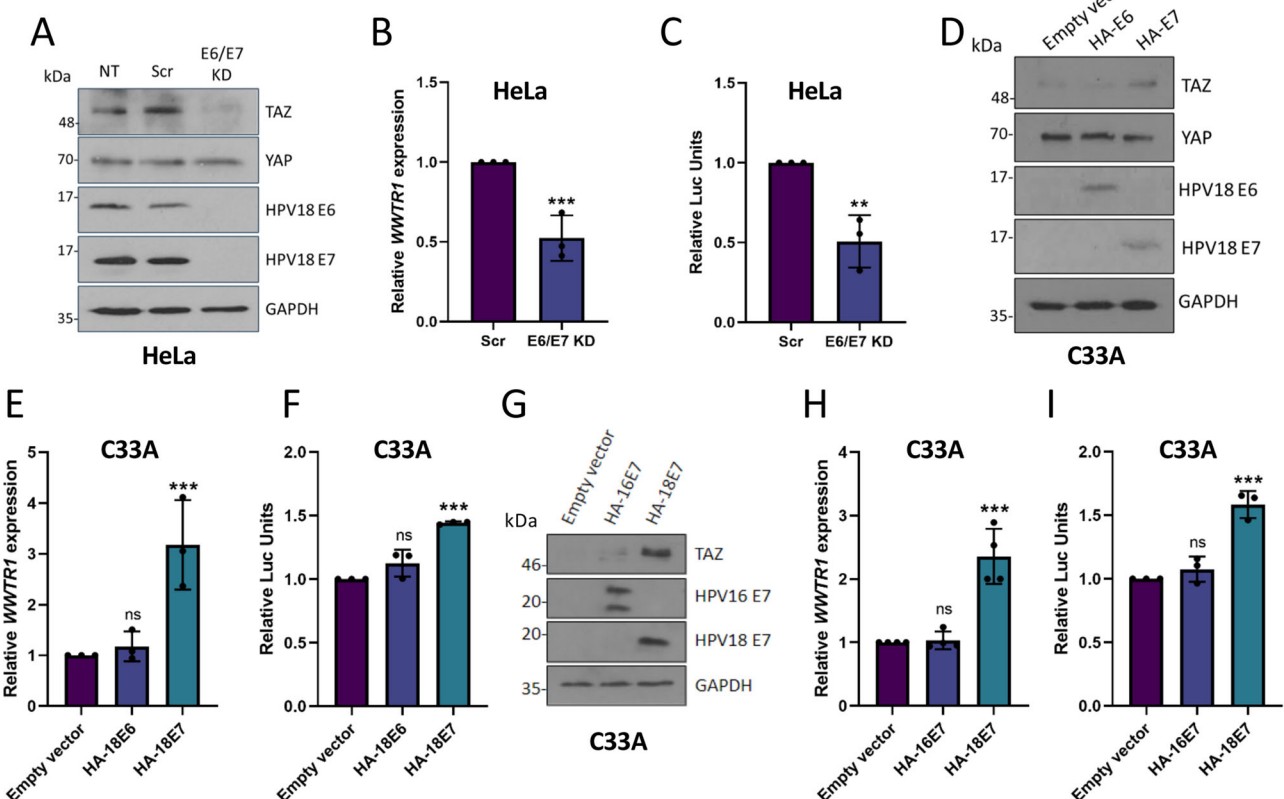

**Fig. 2 | HPV18 E7 upregulates *WWTR1* transcription. A** Representative western blots of HeLa cells after transfection with HPV18 E6/E7 targeting siRNA. Cell lysates were probed for TAZ, YAP, HPV18 E6 and HPV18 E7 expression. GAPDH was used as a loading control (*n* = 4). **B** RT-qPCR analysis of *WWTR1* expression in HeLa cells after transfection with HPV18 E6/E7 targeting siRNA (*n* = 3). *U6* was used as a loading control. **C** *WWTR1*-promoter luciferase assay in HeLa cells after transfection with HPV18 E6/E7 targeting siRNA (*n* = 3). **D** Representative western blot of C33A cell lysates stably expressing either HA-HPV18 E6 or HA-HPV18 E7 probed for TAZ, YAP, HPV18 E6, HPV18 E7 and the loading control GAPDH (*n* = 3). **E** RT-qPCR analysis of *WWTR1* expression in C33A cells stably expressing either HA-HPV18 E6 or HA-

HPV18 E7 (*n* = 3). *U6* was used as a loading control. **F** *WWTR1*-promoter luciferase assay in C33A cells stably expressing either HA-HPV18 E6 or HA-HPV18 E7 (*n* = 3). **G** Representative western blot of C33A cell lysate stably expressing either HA-HPV18 E6 or HA-HPV18 E7 probed for TAZ, YAP, HPV16 E7, HPV18 E7 and GAPDH (*n* = 3). **H** RT-qPCR analysis of *WWTR1* expression in C33A cells stably expressing either HA-HPV16 E7 or HA-HPV18 E7 (*n* = 4). *U6* was used as a loading control. **I** *WWTR1*-promoter luciferase assay in C33A cells stably expressing either HA-HPV16 E7 or HA-HPV18 E7 (*n* = 3). Error bars represent the mean +/- standard deviation of a minimum of three biological repeats. *\*P* < 0.05, *\*\*P* < 0.01, *\*\*\*P* < 0.005 (two-tailed, unpaired Student's *t*-test). Source data are provided as a Source Data file.

in the *WWTR1* promoter (termed site 1 and site 2). We observed enrichment of SP1 binding compared to an IgG isotype control, particularly at the second site. The level of enrichment was significantly greater in the HPV18+ HeLa cells compared to HPV16+ SiHa cells (Fig. 3N). Taken together, these results suggest that HPV18 E7 can enhance *WWTR1*/TAZ expression in part using an ERK1/2-SP1 signalling pathway.

## TAZ promotes proliferation, migration and invasion in HPV18+ cervical cancer cells

We investigated the requirement for TAZ in cervical cancer cells by making use of two small molecule inhibitors that reduce TAZ nuclear localisation and activity. 6079510 is a TAZ inhibitor, whereas the anti-parasitic Ivermectin blocks both YAP and TAZ nuclear entry[24,25]. Immunofluorescence microscopy confirmed the impact of

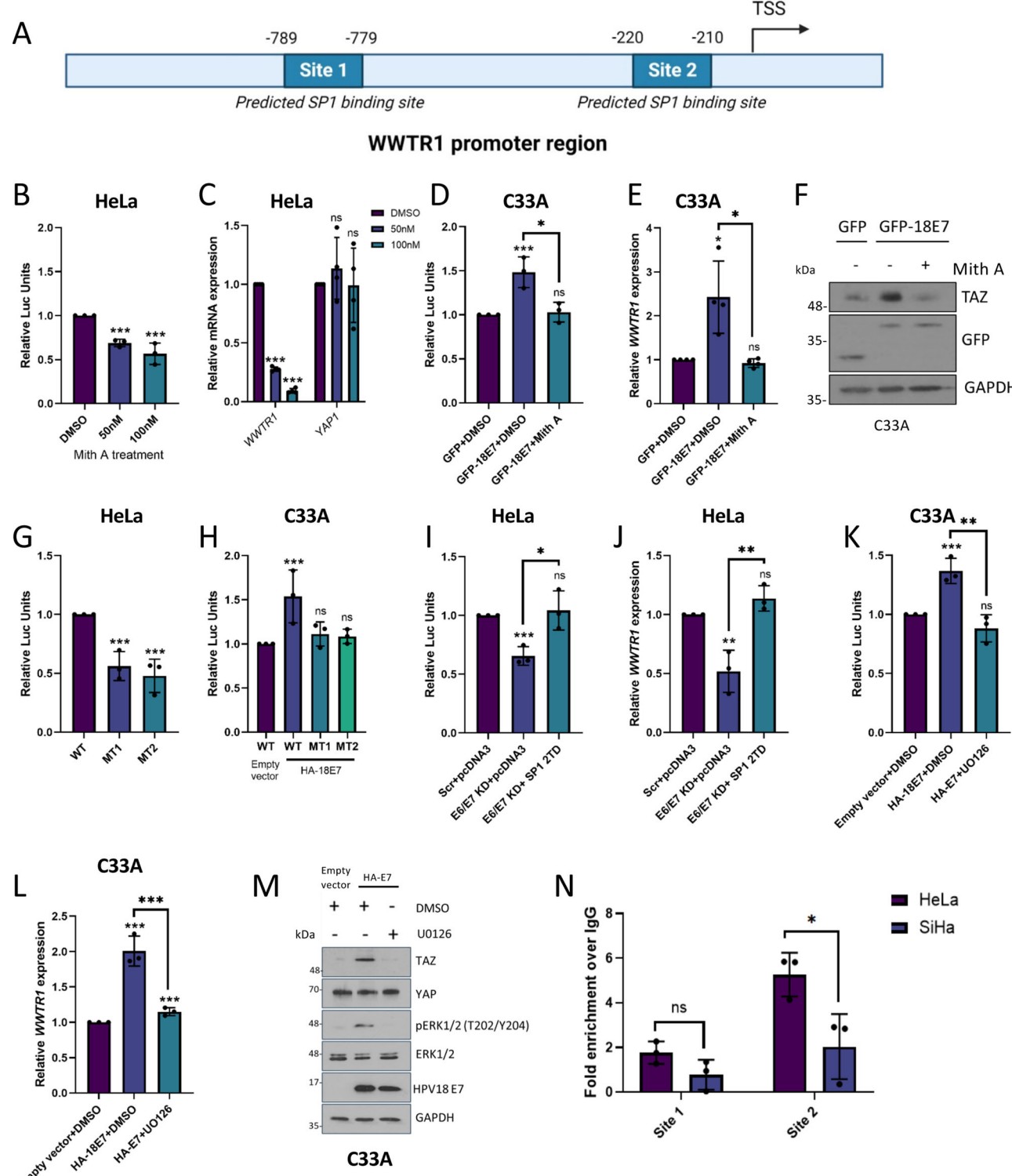

each inhibitor on YAP and TAZ nuclear localisation (Fig. 4A, B – see cells indicated by arrows). 6079510 treatment reduced HeLa cell growth and colony formation (Fig. 4C–E) but had no inhibitory effect in HPV16+ SiHa cells (in fact there was a small positive effect on colony formation) (Fig. 4F–H). In contrast, Ivermectin reduced cell growth and colony formation in both HPV16+ and HPV18+ cell lines (Fig. 4C–H). To confirm our observations, we generated monoclonal HeLa cell lines in which TAZ expression was reduced (but not abolished) using shRNA (Fig. 4I, J). In agreement with our pharmacological data, we observed a loss of proliferation and

colony formation in the two TAZ KD clones (Fig. 4K, M). Knockdown of TAZ led to a small but significant increase in *YAP1* expression, suggesting a degree of compensation between the paralogues (Fig. 4I). No effect of TAZ KD on the expression of the viral onco-proteins was observed (Fig. 4J). Crucially, KD of TAZ expression in another HPV18+ and a HPV45+ cell line (SW756 and MS751 respectively) resulted in a similar reduction in cell growth and colony forming ability (Fig S4).

We were interested in whether TAZ contributes to the trans-forming effects of HPV18 E7. To examine this C33A and HaCaT cells

**Fig. 3 | HPV18 E7-mediated TAZ expression in regulated by the ERK1/2-SP1 signalling axis. A** *WWTR1* promoter schematic showing predicted SP1 binding sites. TSS = transcription start site. Made using BioRENDER.com. **B** *WWTR1*-promoter luciferase assay in HeLa cells following mithramycin A (Mith A) treatment (*n* = 3). **C** RT-qPCR analysis of *WWTR1* and *YAP1* expression in HeLa cells following Mith A treatment (*n* = 4). U6 was used as a loading control. **D** *WWTR1*-promoter luciferase assay in C33A cells following transfection of GFP-HPV18 E7 plus Mith A treatment (*n* = 3). **E** RT-qPCR analysis of *WWTR1* expression in C33A cells following transfection of GFP-HPV18 E7 plus Mith A treatment (*n* = 4). *U6* was used as a loading control. **F** Representative western blots from (**E**) probed for TAZ, GFP and GAPDH (*n* = 3). **G** Luciferase assay in HeLa cells with the *WWTR1*-promoter SP1 binding site mutants (*n* = 3). **H** Luciferase assay in C33A cells stably expressing HA-HPV18 E7 following transfection of the *WWTR1*-promoter SP1 binding site mutants (*n* = 3). **I** *WWTR1*-promoter luciferase assay in HeLa cells transfected with HPV18 E6/E7 targeting siRNA plus SP1 2TD overexpression (*n* = 3). **J** RT-qPCR analysis of

*WWTR1* expression in HeLa cells transfected with HPV18 E6/E7 targeting siRNA plus SP1 2TD overexpression (n = 3). *U6* was used as a loading control. **K** *WWTR1*-promoter luciferase assay in C33A cells stably expressing HA-HPV18 E7 following U0126 treatment (*n* = 3). **L** RT-qPCR analysis of *WWTR1* expression in C33A cells stably expressing HA-HPV18 E7 following U0126 treatment (*n* = 3). *U6* was used as a loading control. **M** Representative western blot of C33A cells stably expressing HA-HPV18 E7 following U0126 treatment probed for TAZ, pERK1/2 (T202/Y204), ERK1/2, HPV18 E7 and GAPDH (*n* = 4). **N** ChIP-qPCR analysis of SP1 binding to the *WWTR1* promoter in HeLa and SiHa cells (*n* = 3). SP1 binding is presented as a fold increase over IgG binding and a gene desert control. Error bars represent the mean +/- standard deviation of a minimum of three biological repeats. *$P < 0.05$, **$P < 0.01$, ***$P < 0.005$ (two-tailed, unpaired Student's *t*-test). Source data are provided as a Source Data file. Figure 3A was created with Biorender.com released under a Creative Commons Attribution-NonCommercial 4.0 International License.

stably expressing HPV18 E7 were treated with the TAZ inhibitor 6079510. As seen in Fig. 4N, O, HPV18 E7 had minimal ability to promote colony formation in the absence of active TAZ in either cell line tested. We conclude that HPV18 E7 protein requires TAZ for its colony forming capability.

Next, we investigated if TAZ contributed to the invasive phenotype of HPV18+ cervical cancer cells using wound healing and Transwell assays, which together demonstrated a reduced migratory and invasive potential in TAZ KD HeLa cells (Fig S5A-D). We hypothesised this may be caused by a disruption of the actin cytoskeleton reorganisation that is associated with cell migration[26]. To investigate this, cells were stained with Rhodamine-Phalloidin, which selectively binds to F-actin allowing protrusions such as filopodia (thin structures consisting of tight bundles of F-actin) to be visualised. TAZ KD significantly reduced both the length of protrusions and the overall number per cell (Fig S5E-G). These data demonstrate that TAZ promotes the proliferative and migratory/invasive phenotype of HPV18+ cervical cancer cells.

### YAP does not compensate for TAZ in HPV18+ cervical cancer cells

To begin to understand their individual roles in cervical cancer, we generated YAP and TAZ KDs using shRNA individually or in combination in HeLa (Fig. 5A, B) or SiHa (Fig. 5F, G) cells and measured the impact on cell growth and colony formation. Loss of either TAZ or YAP was detrimental to HeLa cell growth and colony formation (Fig. 5C-E). Loss of the low levels of endogenous TAZ in SiHa cells caused a small increase in cell growth and colony formation, indicating potential compensatory regulation between paralogues, as has been previously described[27]. In contrast, YAP loss significantly reduced SiHa cell growth (Fig. 5H-J). Similar trends were observed in overexpression experiments, with exogenous YAP and TAZ causing a slight increase in HeLa cell proliferation but only YAP overexpression increasing proliferation in SiHa cells (Fig S6A-D). TAZ overexpression in this instance caused a small, but statistically significant, decrease in proliferation and colony formation (Fig S6E-H).

We next investigated whether YAP overexpression could reverse the defect in cell growth observed in TAZ-depleted HeLa cells. While overexpression of TAZ resulted in a near complete recovery of proliferation and colony formation, YAP overexpression had no significant impact (Fig. 6). We studied the functional determinants necessary for TAZ function in HeLa cells, particularly focussing on the well-characterised interaction with TEAD transcription factors and found that a TEAD-binding-mutant (TBM)[28] was unable to rescue cell growth in the TAZ KD cells (Fig. 6B-D). Similar observations were noted for a TAZ phosphorylation mimic mutant (TAZ S89D), which has reduced nuclear import and thus reduced activity[29] (Fig. 6B-D). Overall, these results suggested that TAZ functions in a manner distinct to YAP in HPV18+ cervical cancer cells to promote cell proliferation, and this role is largely TEAD-dependent.

### TAZ promotes a distinct transcriptional programme to YAP in HPV18+ cervical cancer cells

To understand how YAP and TAZ contribute to cervical cancer cell proliferation, we examined the effect of YAP and TAZ KD on a panel of canonical Hippo pathway-dependent genes (Fig S7). As expected, RT-qPCR analysis showed that expression of each of the four genes was reduced when YAP was depleted. In contrast, their expression increased in TAZ KD HeLa cells (Fig S7A-D), possibly due to the heightened YAP expression previously noted in these cells. This indicates that YAP and TAZ may regulate different target genes in HPV18+ cancer cells. To characterise any TAZ-dependent gene expression programme, we performed RNA-sequencing in YAP and TAZ KD HeLa cells. Integrated analysis of gene expression revealed clear differences in the transcriptional profiles associated with YAP and TAZ (Fig. 7A). Specifically, 79 and 17 genes were up- and down-regulated, respectively, when TAZ was knocked down (Fig. 7B). Gene Ontology analysis of these TAZ regulated genes suggested that the genes suppressed by TAZ expression (upregulated in TAZ KD cells, Data S2) have roles in cell substrate adhesion, epithelial differentiation and the immune response, all of which are known to play key roles in cancer [e.g[30,31].] (Data S3). In contrast, genes shown to be positively regulated by TAZ (downregulated in TAZ KD) are typically involved in proliferation and invasion [e.g[32].], although no GO terms were significantly over-represented within this gene list (Data S4).

To understand how TAZ influences cervical cancer, we focused upon *TOGARAM2* (a.k.a. *FAM179A*) for further analysis, as RNA-sequencing data indicated that it was significantly repressed in TAZ KD cells compared with negative control (A1; LogFC = − 4.04, adjusted *P*-value = 0.0000697. A2; LogFC = −6.93, adjusted *P*-value = 0.000242), whilst not significantly altered in YAP KD cells (B1; LogFC = −0.484, adjusted *P*-value = 0.228. B2; LogFC = 0.415, adjusted *P*-value = 0.243) (Fig. 7C, D). The function of TOGARAM2 is not known but it has been identified to be among genes significantly overexpressed in relapsing ALK-positive anaplastic large cell lymphoma[33]. RT-qPCR confirmed that *TOGARAM2* expression was inhibited when TAZ, but not YAP, expression was depleted by shRNA (Fig S8A). *TOGARAM2* was confirmed to be a TAZ-dependent gene in additional HPV18+ (SW756) and HPV18-like (HPV45 + ) (MS751) cells (Fig S8B, C). *TOGARAM2* expression was reduced in the presence of the TAZ inhibitor 6079510, or Ivermectin (Fig S8D) and was increased in cells expressing HPV18 E7 in a TAZ-dependent manner (Fig S8E). Finally, verteporfin treatment, which inhibits the interaction of TAZ with the co-transcriptional regulator TEAD, also repressed *TOGARAM2* expression (Fig S8F).

### TOGARAM2 expression is upregulated in HPV18+ cervical cancers and is associated with poor survival

To begin to understand the functional association between TAZ and TOGARAM2 in cervical cancer, we first investigated *TOGARAM2's* expression in a panel of cell lines. This showed *TOGARAM2* expression

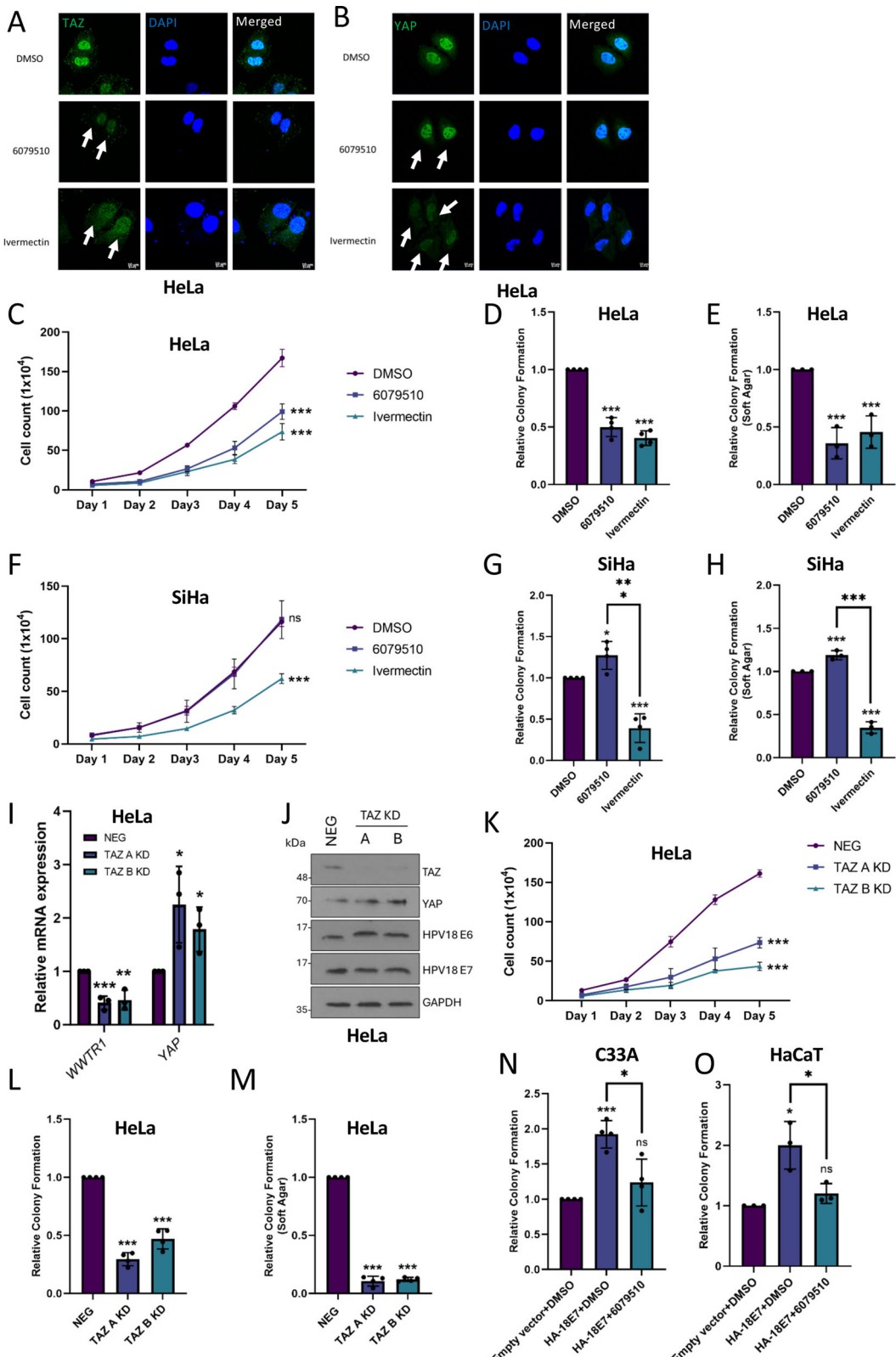

was specifically upregulated in the HPV18+ and HPV45+, but not HPV16+ cell lines, like TAZ (Fig. 7E). Although we were not able to detect TOGARAM2 protein expression due to a lack of available validated antibody reagents, we found that *TOGARAM2* mRNA expression was significantly increased in the high-grade cervical lesions of HPV18+ patients, but not HPV16+ lesions, and this again positively correlated with *WWTR1* expression (Fig. 7F, G). Finally, we investigated

the clinical relevance of *TOGARAM2* expression in cervical cancer patients. Similarly, to the scenario observed for *WWTR1*, high *TOGARAM2* expression correlated with reduced progression-free survival specifically in HPV18+ cervical cancer patients, but this did not reach statistical significance (Fig. 7H, I). Our data support that *TOGARAM2* is a TAZ-dependent gene that may function to promote cervical cancer progression.

**Fig. 4 | Inhibition of TAZ reduces HPV18+ cervical cancer cell proliferation.** **A** Immunofluorescence microscopy analysis of TAZ localisation (green) in HeLa cells treated with 6079510 or Ivermectin. DAPI stained nuclei (blue). Arrows point to nuclear TAZ. Scale bar 10 μm ($n = 3$). **B** Immunofluorescence microscopy analysis of nuclear YAP (green) in HeLa cells treated with 6079510 or Ivermectin. DAPI stained nuclei (blue). Arrows point to nuclear YAP. Scale bar 10 μm ($n = 3$). **C** Growth curve analysis of HeLa cells treated with 6079510 or Ivermectin ($n = 3$). **D** Colony formation assay of HeLa cells treated with 6079510 or Ivermectin ($n = 4$). **E** Soft agar assay of HeLa cells treated with 6079510 or Ivermectin ($n = 3$). **F** Growth curve analysis of SiHa cells treated with 6079510 or Ivermectin ($n = 3$). **G** Colony formation assay of SiHa cells treated with 6079510 or Ivermectin ($n = 4$). **H** Soft agar assay of SiHa cells treated with 6079510 or Ivermectin ($n = 3$). **I** RT-qPCR analysis of *WWTR1* or *YAP1* expression in TAZ KD HeLa cells ($n = 3$). *U6* was used as a loading control. **J** Representative western blot of TAZ KD HeLa cells probed for TAZ, HPV18 E6, HPV18 E7 and GAPDH ($n = 3$). **K** Growth curve analysis of TAZ KD HeLa cells ($n = 3$). **L** Colony formation assay in TAZ KD HeLa cells ($n = 4$). **M** Soft agar assay in TAZ KD HeLa cells ($n = 4$). **N** Colony formation assay in C33A cells stably expressing HA-HPV18 E7 plus 6079510 treatment ($n = 4$). **O** Colony formation assay in HaCaT cells stably expressing HA-HPV18 E7 plus 6079510 treatment ($n = 3$). Error bars represent the mean +/- standard deviation of a minimum of three biological repeats. *$P < 0.05$, **$P < 0.01$, ***$P < 0.005$ (two-tailed, unpaired Student's *t*-test). Source data are provided as a Source Data file.

## TOGARAM2 is essential for TAZ-mediated migration and invasion of HPV18+ cervical cancer cells

To determine the importance of TAZ-mediated upregulation of TOGARAM2 in HPV18+ cervical cancer, we generated polyclonal TOGARAM2 KD HeLa cell lines using two different shRNA (Fig. 8A). The TOGARAM2 KD cells showed significantly decreased cell growth and colony formation (Fig. 8B–D). Furthermore, we also observed a significant decrease in the number of protrusions per cell and their length (Fig. 8E–G). When TOGARAM2 was further overexpressed in HeLa cells, we saw a consistently small, but significant, increase in cell growth and colony formation (Fig S9A–D). When over-expressed in C33A cells, which express low levels of TOGARAM2, we observed a much greater increase in cell growth, colony formation, migration and invasion (Fig S9 E–I). In contrast, knockdown of the already low levels of TOGARAM2 in SiHa cells resulted in no significant changes in cell growth, while knockdown in another HPV18+ cell line (SW756) led to a similar reduction in growth as observed in the HeLa cells (Fig S10 A–D).

To understand whether TOGARAM2 is important for the TAZ−dependent phenotype in cervical cancer cells, we overexpressed TOGARAM2 in TAZ KD HeLa cells (Fig. 9A). Whilst reintroduction of TOGARAM2 into these cells caused a partial rescue in cell growth and colony formation (Fig. 9B–D), we observed a near complete restoration of migratory and invasive capacity of the TAZ KD HeLa cells (Fig. 9E, F). This correlated with a rescue in the number and length of cellular protrusions observed (Fig. 9G–I). Taken together, our data demonstrate that TOGARAM2 mediates aspects of the oncogenic functions of TAZ, particularly associated with cell migration and invasion and suggests that TOGARAM2 is a potential oncogene in HPV18+ cervical cancer cells.

## Discussion

With no current HPV-specific treatments for cervical cancer available, understanding HPV-induced transformation is of vital importance to enable the development of therapeutics. Additionally, identifying specific differences between HPV16+ and HPV18+ disease is critical, as HPV18 infection is linked with higher mortality rates and a higher chance of metastasis[34]. In this study, we demonstrate that TAZ is specifically upregulated in HPV18+ cervical cancers where it is a critical driver of cell proliferation, migration and invasion.

Although past studies have highlighted the Hippo signalling pathway as an important regulator in cervical cancer[10], work has primarily focused on the YAP transcription factor whilst the role of TAZ has remained elusive[15]. Here, we demonstrate that TAZ functions as an oncogene in HPV18+ (and HPV18-like), but not HPV16+ cervical cancer, highlighting a key difference between HPV16- and HPV18-driven cervical disease. We used a combination of patient samples and cell lines to demonstrate that TAZ expression correlates with cervical disease progression and is specifically upregulated in HPV18+ cervical tissue and cell lines. Furthermore, we demonstrate that high TAZ expression correlates with a worse overall survival in HPV18+ cervical cancer patients, but not in those without HPV infection or infected with non-HPV18 types. Given our results observing a similar phenotype in a HPV18-like, HPV45+ cell line to that of HPV18+ cell lines, future work should aim to expand this and scrutinise the phenotype of TAZ across a broader range of HPV types with key roles in HPV-driven cancer development.

Our study indicates that TAZ transcription can be upregulated by the major HPV E7 oncoprotein by a mechanism requiring MAPK signalling and the SP1 transcription factor. Interestingly, we did not observe increased TAZ expression in CIN1 samples, which are likely to still maintain productively replicating virus, suggesting that TAZ may not play a role during HPV infection, though further investigation is needed to confirm this. This is important, as a recent study has shown that YAP helps to maintain a basal cell identity in HPV infected keratinocytes[11], suggesting potential differences in requirement for YAP and TAZ between productive virus infection and disease. Additionally, this study highlights the complex and poorly understood differences between HPV16 and HPV18, as we found that HPV16 E7 was unable to induce TAZ expression, despite previous studies demonstrating its ability to regulate SP1 activity[20]. Clearly, further work will be essential to unravel precise pathways contributing to SP1 activation potentially beyond ERK1/2 and the differences in the SP1 transcriptomes in HPV16+ and HPV18+ cancers. Likewise, although we showed a degree of SP1 occupation of the *WWTR1* promoter in HPV18+ cells, there was some SP1 binding to the *WWTR1* promoter in HPV16+ cells, indicating that additional mechanisms and factors might be required. As such, further investigation is needed to elucidate how the promoter is controlled. This could include examining histone markers or transcriptional co-factors to gain an understanding of the control of this promoter, furthering not only our understanding of HPV biology but also TAZ biology in disease.

Although YAP activity and expression has been demonstrated to be upregulated in HPV+ cancers, knockdown of HPV oncogene levels in HPV+ cervical cancer cell lines had no observable effect on YAP expression in our hands. This suggests that HPV+ cancers may have lost their dependence on the viral oncogenes to maintain high levels of YAP, despite requiring continued YAP activity. Further research is required to understand this and what cellular changes occur in HPV+ cancers that may be the cause of this phenomenon.

We found that inhibition of TAZ in HPV18+ cell lines, either through pharmacological inhibition or shRNA-mediated knockdown impeded proliferation, migration and invasion. We also demonstrated that TAZ acts in an independent capacity to YAP in this context, controlling proliferation in a non-redundant manner, illustrated by the lack of rescue for TAZ KD cells following YAP overexpression. This further supports a paradigm where YAP and TAZ can play distinct roles in cancer, and future research should aim to understand them further as separate entities. Moreover, we found that TAZ overexpression was in fact an impediment to the growth of HPV16+ cell lines and was seen to positively impact patient survival. This aligns with growing reports of YAP and TAZ serving tumour suppressive functions in certain cellular contexts. For instance, in multiple myeloma TAZ expression positively correlates with disease outcome, and TAZ is frequently silenced by DNA methylation[35], as it has been shown to inhibit proliferative targets such as c-Myc[35]. It will be of interest to understand if TAZ has specific

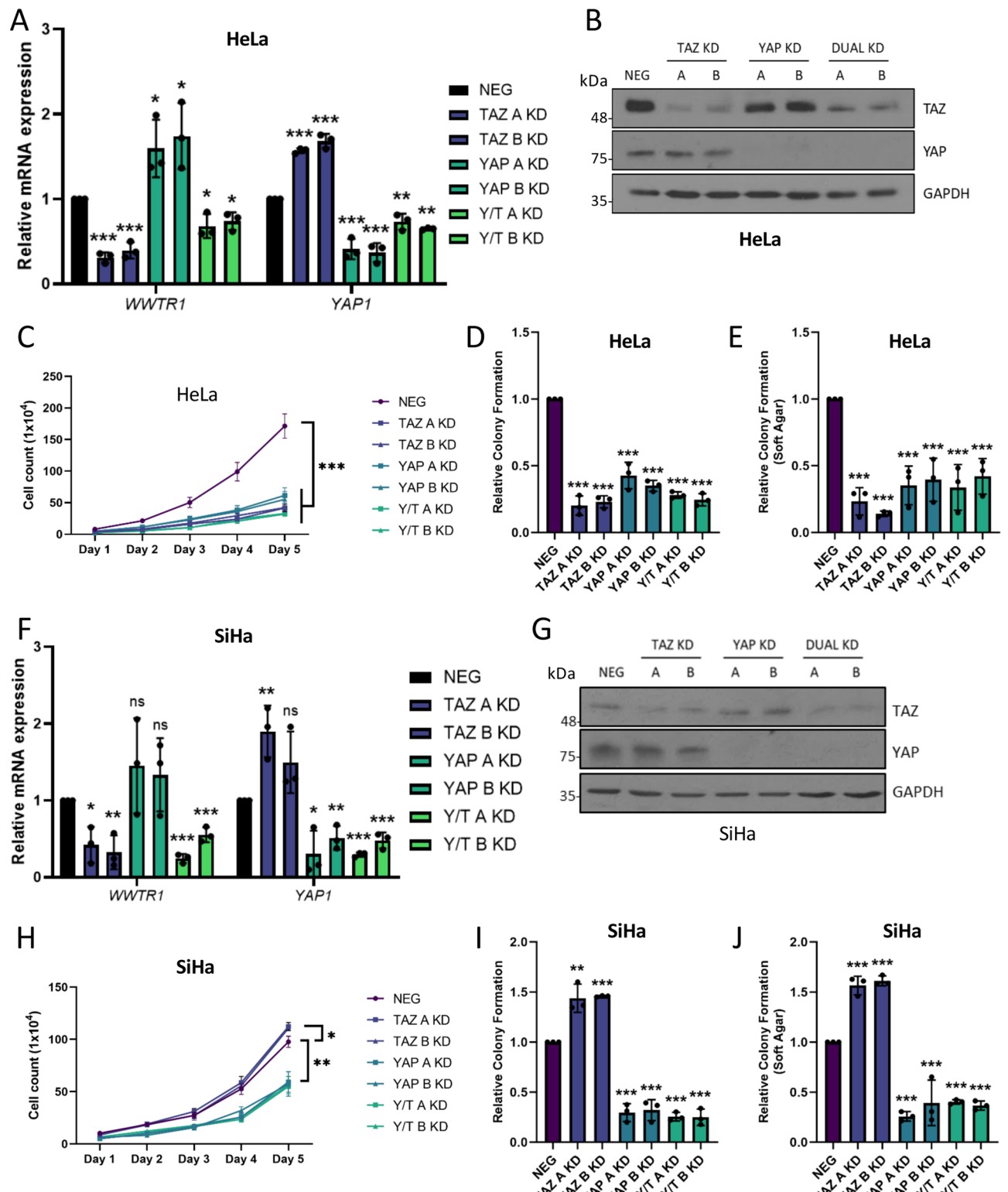

**Fig. 5 | YAP and TAZ are oncogenic in HPV18 + , but not HPV16+ cervical cancer cell lines. A** RT-qPCR analysis of *WWTR1* and *YAP1* expression in TAZ, YAP or YAP/TAZ (Y/T) KD HeLa cells (*n* = 3). *U6* was used as a loading control. **B** Representative western blots of TAZ, YAP and loading control GAPDH from TAZ, YAP or YAP/TAZ (Y/T) KD HeLa cell lysates (*n* = 3). **C** Growth curve analysis of TAZ, YAP or YAP/TAZ (Y/T) KD HeLa cells (*n* = 3). **D** Colony formation assay of TAZ, YAP or YAP/TAZ (Y/T) KD HeLa cells (*n* = 3). **E** Soft agar assay of TAZ, YAP or YAP/TAZ (Y/T) KD HeLa cells (*n* = 3). **F** RT-qPCR analysis of *WWTR1* or *YAP1* expression in TAZ, YAP or YAP/TAZ (Y/T) KD SiHa cells (*n* = 3). *U6* was used as a loading control. **G** Representative western blot of YAP, TAZ and GAPDH from TAZ, YAP or YAP/ TAZ (Y/T) KD SiHa cell lysates (*n* = 3). **H** Growth curve analysis of TAZ, YAP or YAP/TAZ (Y/T) KD SiHa cells (*n* = 3). **I** Colony formation assay of TAZ, YAP or YAP/TAZ (Y/T) KD SiHa cells (*n* = 3). **J** Soft agar assay of TAZ, YAP or YAP/TAZ (Y/T) KD SiHa cells (*n* = 3). Error bars represent the mean +/- standard deviation of a minimum of three biological repeats. *$P < 0.05$, **$P < 0.01$, ***$P < 0.005$ (two-tailed, unpaired Student's *t*-test). Source data are provided as a Source Data file.

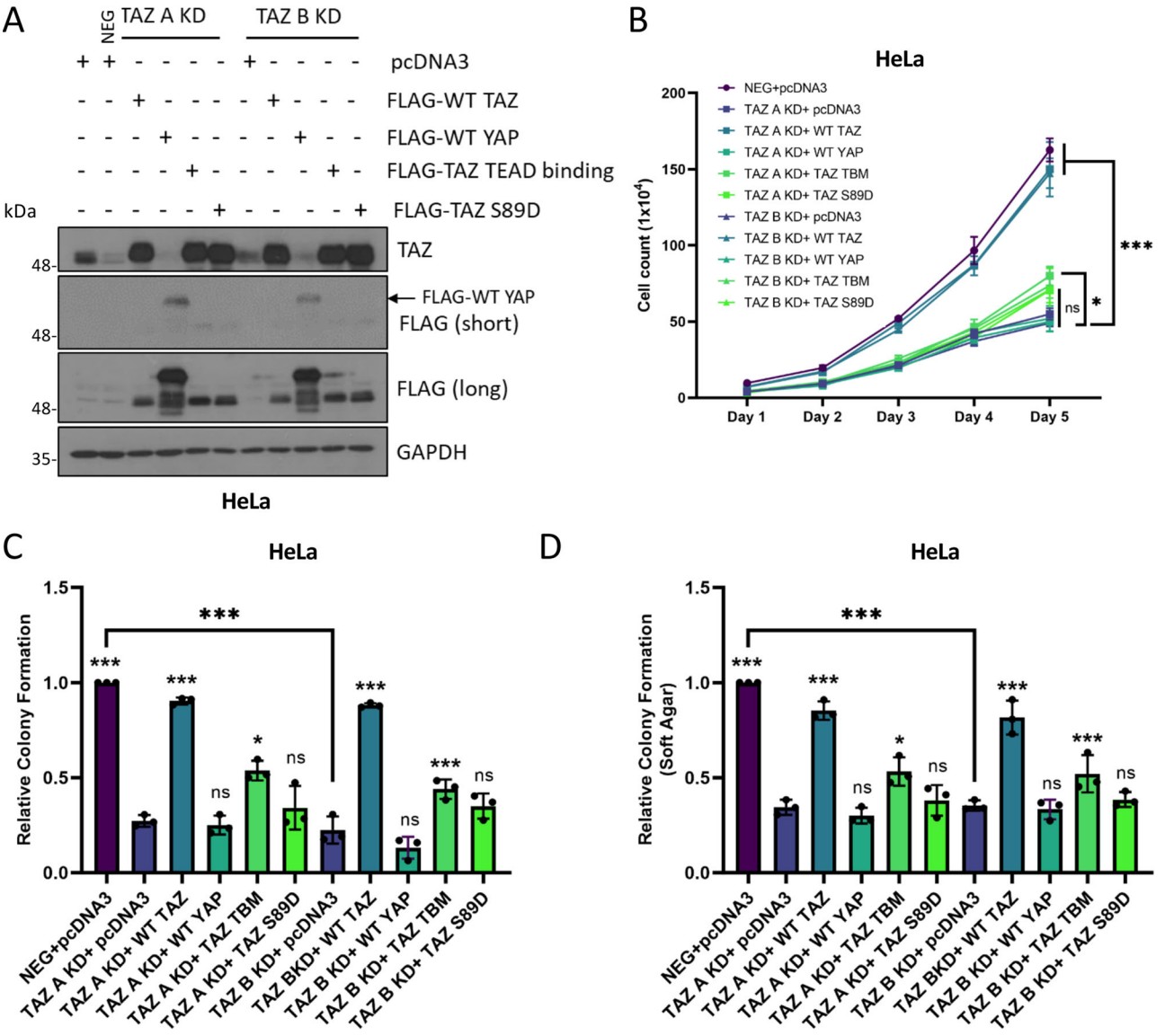

**Fig. 6 | YAP or a TAZ mutant defective in TEAD binding cannot rescue the loss of proliferation in TAZ KD HeLa cells. A** Representative western blots from TAZ KD HeLa cell lysates transfected with pcDNA3, FLAG-TAZ, FLAG-YAP, FLAG-TAZ TEAD binding mutant and FLAG-TAZ S89D, probed for TAZ, FLAG and the loading control GAPDH (*n* = 3). **B** Growth curve analysis of TAZ KD HeLa cell lysates transfected with either pcDNA3, FLAG-TAZ, FLAG-YAP, FLAG-TAZ TEAD binding mutant and FLAG-TAZ S89D (*n* = 3). Colony formation assay (**C**) or soft agar assay

(**D**) of TAZ KD HeLa cells transfected with pcDNA3, FLAG-TAZ, FLAG-YAP, FLAG-TAZ TEAD binding mutant or FLAG-TAZ S89D (*n* = 3). Statistical analysis of each condition is compared to each TAZ KD cell line. Error bars represent the mean +/- standard deviation of a minimum of three biological repeats. *$P$ < 0.05, **$P$ < 0.01, ***$P$ < 0.005 (two-tailed, unpaired Student's *t*-test). Source data are provided as a Source Data file.

targets in HPV16+ cells. In HPV18+ cells, we suggest that YAP and TAZ regulate distinct transcription profiles, as revealed by RNA sequencing, suggesting YAP and TAZ control cancer hallmarks through different target genes. One gene which we found to be specifically regulated by TAZ was *TOGARAM2* which we identify as a previously uncharacterised effector of TAZ-mediated proliferation, migration and invasion. Additionally, we show that TOGARAM2 is a potential regulator of cellular architecture, with TOGARAM2 knockdown leading to a loss of cellular protrusions. Interestingly, TOGARAM2 reintroduction was able to recover migration and invasion following TAZ knockdown. This study identifies a contribution of TOGARAM2 in the key cancerous phenotypes; proliferation, migration and invasion. Therefore, we suggest TOGARAM2 is an oncogene and a potential metastatic factor. As TAZ is frequently linked to aggressive and metastatic phenotypes, it is possible TOGARAM2 serves as a TAZ target in other non-HPV driven cancers, and further studies will be

required to confirm this. Future work should also aim to clarify how TOGARAM2 contributes towards carcinogenesis by identifying TOGARAM2 binding partners. As TOGARAM2 contains two Tumour Overexpressed Gene (TOG) domains, it is likely these domains mediate interactions with the cellular cytoskeleton, similar to other TOG-domain containing proteins, explaining the phenotype we observed regarding cellular migration, invasion and cellular protrusions[36].

Although this study has delineated the contribution of a single TAZ-dependent gene to cervical cancer hallmarks, future studies should aim to characterise more of these genes to broaden our understanding of the critical role of TAZ in cancer biology. Interestingly, RNA-sequencing data revealed that TAZ suppresses the transcription of multiple genes. Gene ontology analysis predicts that several of these targets have putative tumour suppressive functions. It will be important to both understand how TAZ mediates their

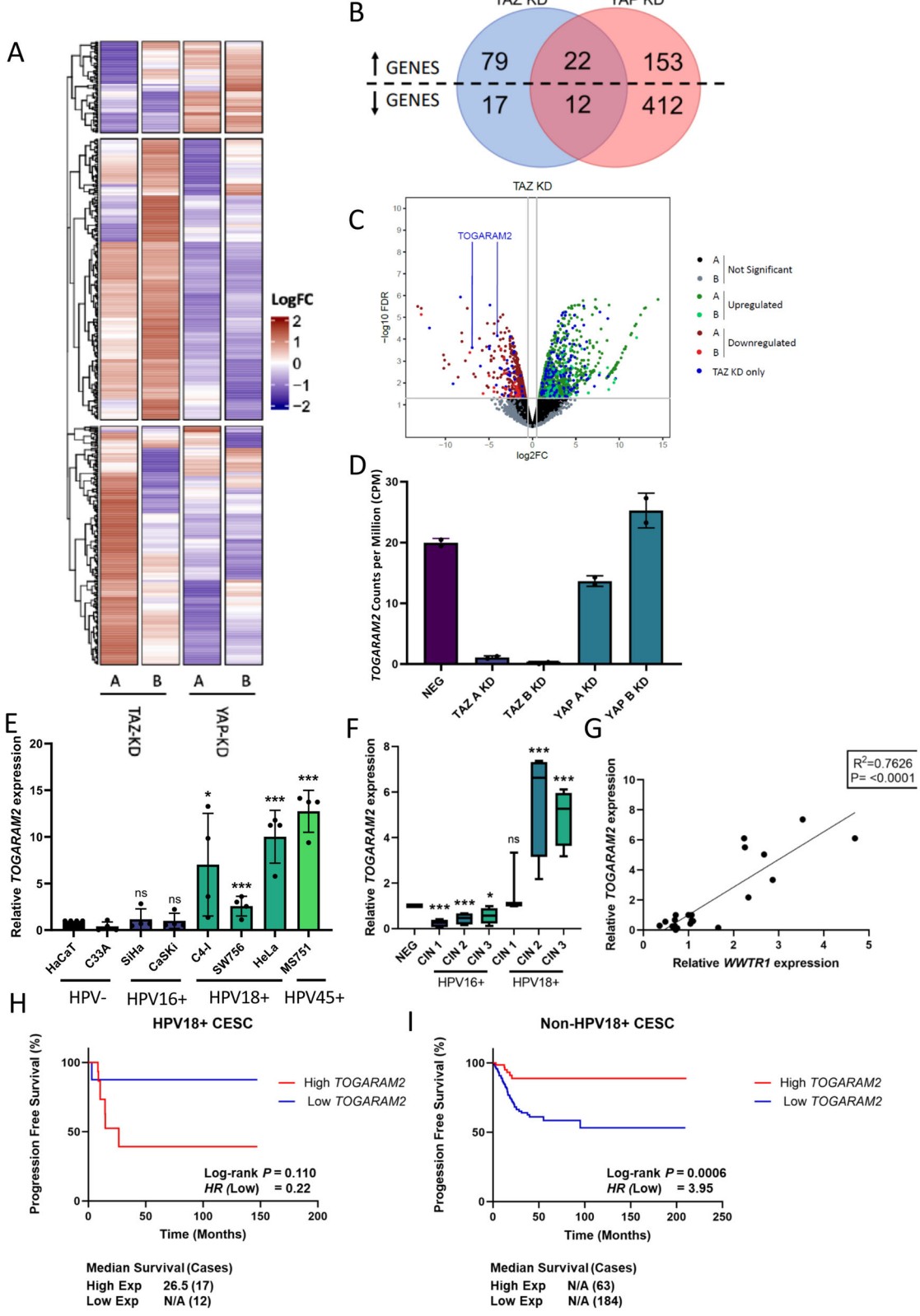

repression and to decipher their role in cell biology to understand why their suppression is required in the HPV18+ cervical cancers. More broadly, these findings further highlight differences between YAP and TAZ function and remind us that we still do not completely understand their functions.

To summarise, here we present a study of the role of TAZ in HPV-driven cervical cancer. We show that TAZ is specifically upregulated by HPV18 E7 in an SP1-dependent manner and plays an oncogenic role in HPV18+ cervical cancer cell lines. Furthermore, we show that YAP and TAZ play non-redundant roles in HPV18+ cervical cancer cells by promoting distinct transcription profiles. We identified TOGARAM2 as an example of a TAZ-dependent gene and our work demonstrates TOGARAM2 as an oncogene and potential metastatic factor (Fig. 10).

**Fig. 7 | RNA-seq analysis demonstrates distinct transcriptional profiles controlled by YAP and TAZ and identify TOGARAM2 as a TAZ-dependent gene.**
**A** LogFC Heatmap of differentially expressed genes in TAZ KD (A and B KD cell lines) YAP KD (A and B KD cell lines) HeLa cell lines ($n = 2$). LogFC are scaled row-wise, where red and blue indicate a LogFC higher and lower than the row-wise statistical mean, respectively. **B** Venn diagram of DEGs upregulated and down-regulated in TAZ KD (A and B) compared with NEG and YAP KD (A and B) compared with NEG, including overlapping DEGs in each comparison contrast. **C** Volcano plot of TAZ KD RNA-sequencing in HeLa cells defined significant at adjusted *P*-value < 0.05 and logFC threshold at 0.5. Non-significantly altered genes are highlighted black and grey in A1 and A2 contrasts, respectively. Significantly upregulated genes are highlighted dark green and light green in A1 and A2 contrasts, respectively. Significantly downregulated genes are highlighted dark red and light red in A1 and A2 contrasts, respectively. DEGs significantly altered in TAZ KD only are highlighted blue. **D** Counts per million of *TOGARAM2* expression in

RNA-sequencing of NEG, TAZ KD (A and B KD cell lines) and YAP KD (A and B KD cell lines) ($n = 2$). **E** RT-qPCR analysis of *TOGARAM2* expression in HPV-, HPV16+ or HPV18+ cell lines ($n = 4$). *U6* transcript levels were used as a loading control.
**F** RT-qPCR analysis of *TOGARAM2* expression in negative, HPV16 or HPV18+ patient cervix liquid cytology samples from CIN grades ($n = 4$ from each grade). *U6* was used as a loading control. Maxima and minima whiskers represent the highest and lowest values, respectively. Bounds of boxes are the 75th and 25th percentiles and the centre represents the median. **G** Graph showing correlation between *TOGARAM2* and *WWTR1* expression from **C**. The correlation coefficient (r) was calculated using Pearson correlation analysis. **H**, **I** Kaplan–Meier curves showing overall survival in cervical cancer stratified by HPV18+ or non-HPV18+. Survival was compared using the log-rank test. Errors represent the mean +/- standard deviation of a minimum of three biological repeats. *$P < 0.05$, **$P < 0.01$, ***$P < 0.005$ (two-tailed, unpaired Student's t-test). Source data are provided as a Source Data file.

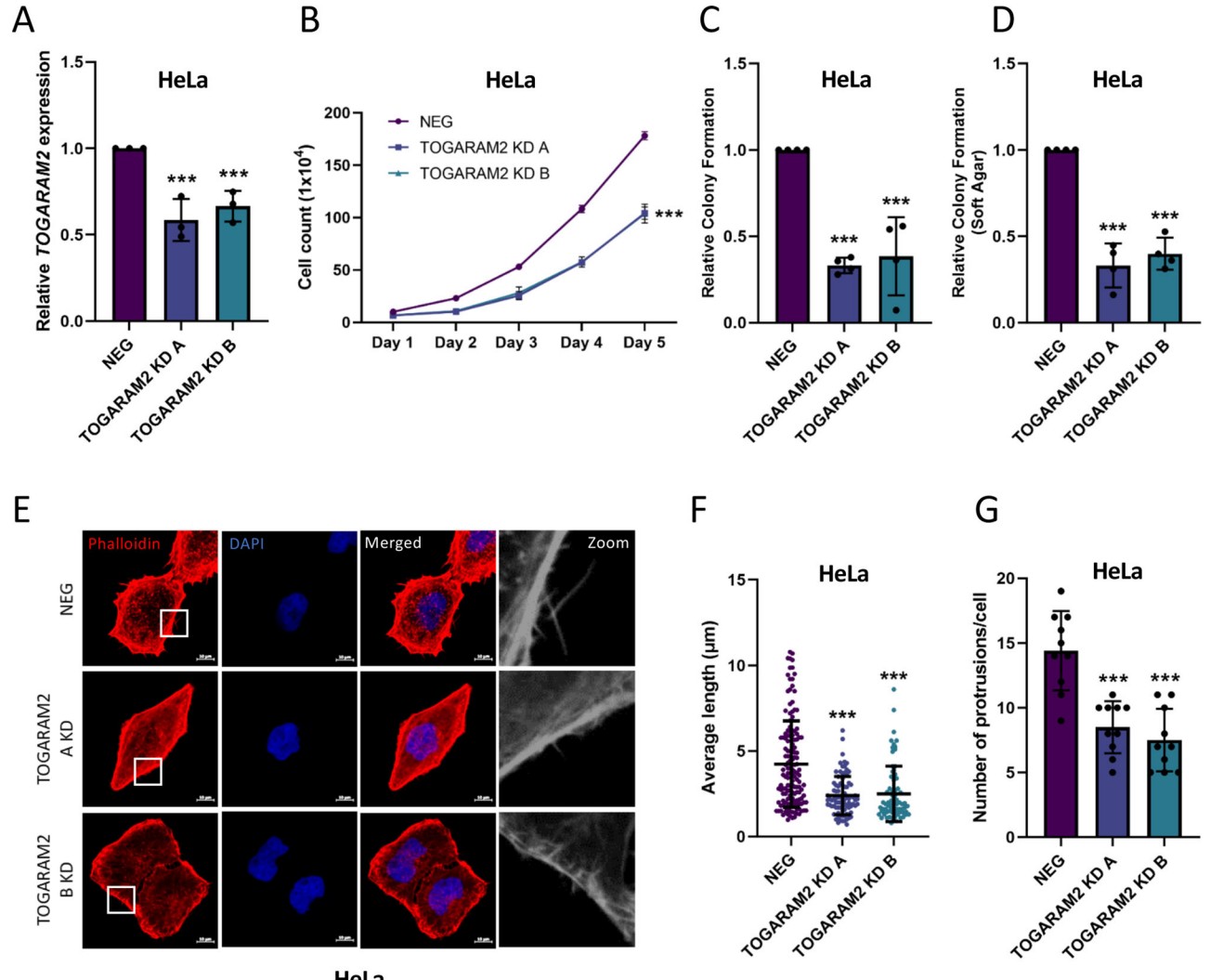

**Fig. 8 | TOGARAM2 is pro-oncogenic in HPV18+ cervical cancer cells. A** RT-qPCR analysis of *TOGARAM2* expression in TOGARAM2 KD HeLa cells ($n = 3$). *U6* was used as a loading control. **B** Growth curve analysis of TOGARAM2 KD HeLa cells ($n = 3$). **C** Colony formation assays in TOGARAM2 KD HeLa cells ($n = 4$). **D** Soft agar assays in TOGARAM2 KD HeLa cells ($n = 4$). **E** Immunofluorescence microscopy analysis of Rhodamine-Phalloidin stained (red) TOGARAM2 KD HeLa cells. DAPI stained nuclei

(blue). Scale bar 10 μm. **F** Measurement of filopodia length in TOGARAM2 KD HeLa cells ($n = 10$). **G** Average number of filopodia per cell in TOGARAM2 KD HeLa cells ($n = 10$). Error bars represent the mean +/- standard deviation of a minimum of three biological repeats. *$P < 0.05$, **$P < 0.01$, ***$P < 0.005$ (two-tailed, unpaired Student's *t*-test). Source data are provided as a Source Data file.

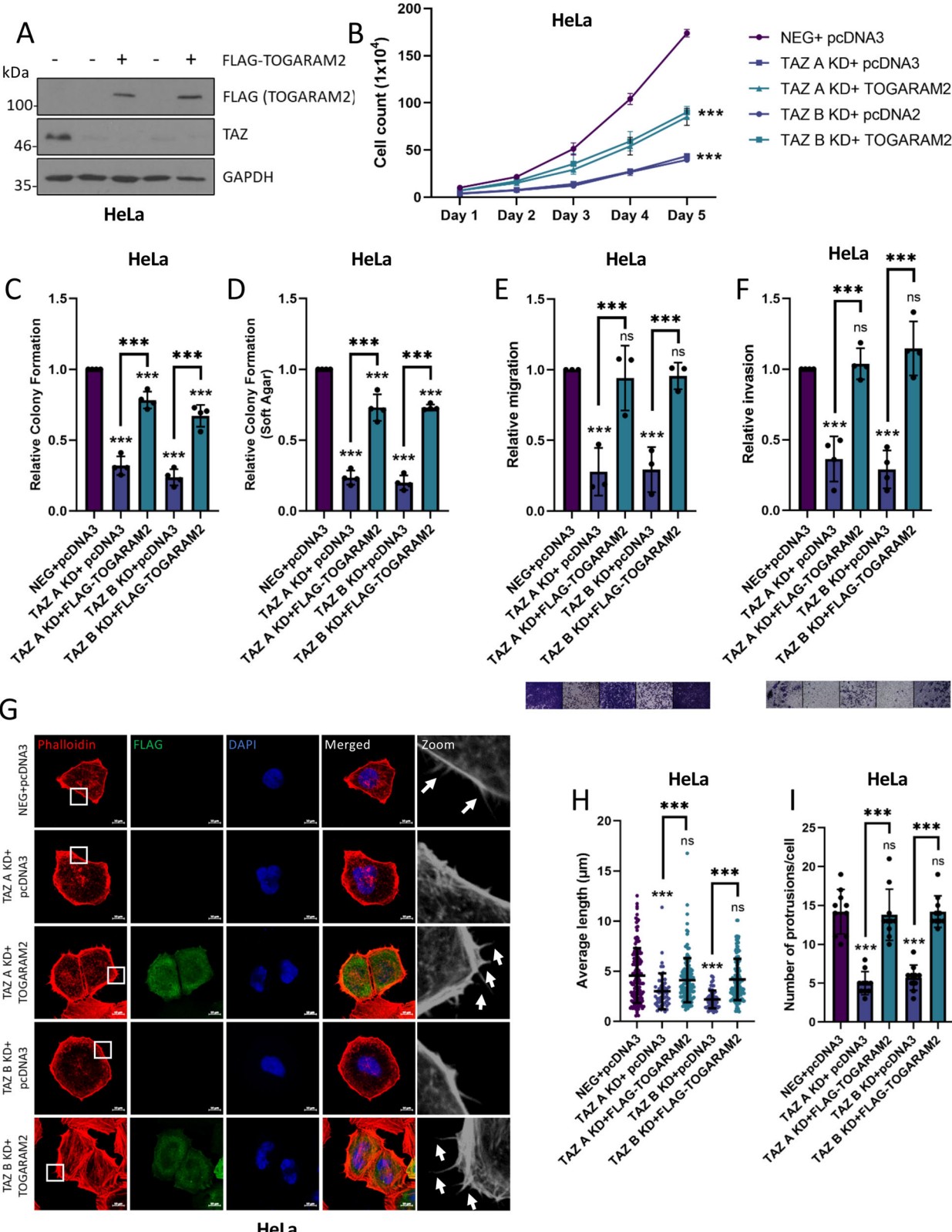

## Methods

### Ethics statement

All studies comply with all relevant ethical regulations. Release and use of cervical cytology material (which were surplus to diagnostic requirements and associated with informed consent) is covered by the research tissue bank approval held by the Scottish HPV archive which comes under the National Research for Scotland BioResource [research tissue bank approval REC Ref:20/ES/0061]. Specific samples were available through an application to the Archive Steering Committee (App Ref 0034).

### TCGA data analysis

Patient progression-free survival data were downloaded from the Supplemental Information of a pan-cancer clinical study[37]. The clinical

**Fig. 9 | TOGARAM2 contributes towards TAZ-driven oncogenicity.**
**A** Representative western blots of TAZ KD HeLa cell lysates transfected with pcDNA3 or FLAG-TOGARAM2 probed for TAZ, FLAG and GAPDH loading control ($n = 3$).
**B** Growth curve analysis of TAZ KD HeLa cell lysates transfected with pcDNA3 or FLAG-TOGARAM2 ($n = 3$). **C** Colony formation assay of TAZ KD HeLa cell lysates transfected with pcDNA3 or FLAG-TOGARAM2 ($n = 4$). **D** Soft agar assay of TAZ KD HeLa cell lysates transfected with pcDNA3 or FLAG-TOGARAM2 ($n = 4$). **E** Transwell migration assay of TAZ KD HeLa cells transfected with pcDNA3 or FLAG-TOGARAM2 ($n = 3$). **F** Cell Invasion assay of TAZ KD HeLa cells transfected with pcDNA3 or FLAG-

TOGARAM2 ($n = 4$). **G** Immunofluorescence microscopy analysis of Rhodamine-Phalloidin stained (red) TAZ KD HeLa cells transfected with pcDNA3 or FLAG-TOGARAM2. DAPI stained nuclei (blue). Scale bar 10 μm. **H** Measurement of filopodia length in TAZ KD HeLa cells transfected with pcDNA3 or FLAG-TOGARAM2 ($n = 10$). **I** Average number of filopodia per cell in TAZ KD HeLa cells transfected with pcDNA3 or FLAG-TOGARAM2 ($n = 10$). Error bars represent the mean +/- standard deviation of a minimum of three biological repeats. *$P < 0.05$, **$P < 0.01$, ***$P < 0.005$ (two-tailed, unpaired Student's $t$-test). Source data are provided as a Source Data file.

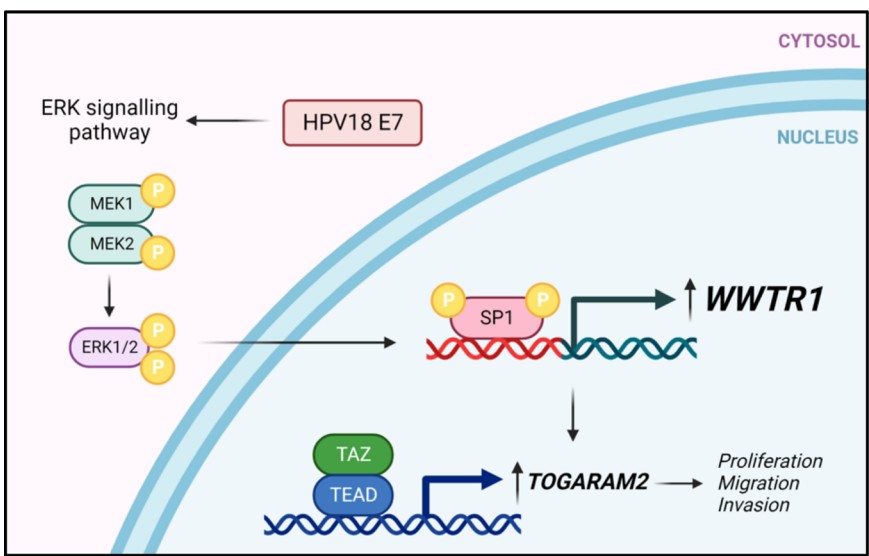

**Fig. 10 | HPV18 E7-mediated upregulation of TAZ expression and activity in cervical cancer.** The HPV18 E7 oncoprotein activates the ERK1/2 MAPK pathway resulting in activation of the SP1 transcription factor, which drives the increased expression of TAZ. Subsequently TAZ is responsible for a transcriptional programme necessary for the increased proliferation, migration and invasion observed in HPV18-positive cancer cells and this is partially dependent on the actions of the TAZ target TOGARAM2. Figure 10 was created with Biorender.com released under a Creative Commons Attribution-NonCommercial 4.0 International License.

survival endpoints for patients were progression-free survival (PFS), which is the time from the date of diagnosis until tumour progression or death, whichever occurs first. The PFS curves were obtained using Kaplan–Meier method with expression cut off determined using KMPlotter[38] and were compared using the log-rank test. The Cox proportional hazards model was used to estimate Hazard Ratios (HRs) with 95% Confidence Intervals (CIs).

### Cervical cytology samples
De-identified cervical cytology samples were obtained from the Scottish HPV Archive (http://www.shine/mvm.ed.ac.uk/archive.shtml), a biobank of over 20,000 samples designed to facilitate HPV-associated research. RNA was extracted from samples using Trizol (ThermoFisher Scientific, USA)[39] and analysed as described.

### Cell culture
HaCaT (immortalized human keratinocytes) were provided by Dr Martin Stacey, University of Leeds. C33A (HPV negative cervical squamous carcinoma cells) (HTB-31), SiHa (HPV16+ cervical squamous carcinoma cells) (HTB-35), CaSKi (HPV16+ cervical squamous carcinoma cells) (CRM-CRL-1550), SW756 (HPV18+ cervical squamous carcinoma cells) (CRL-3584), C4-I (HPV18+ squamous carcinoma of the uterine cervix) (CRL-1594), HeLa (HPV18+ cervical epithelial adenocarcinoma) (CRM-CCL-2), and MS751 (HPV45+ cervical squamous cell carcinoma derived from lymph node) (HTB-34) cell lines were obtained from ATCC. MS751 were grown in Eagle's Minimum Essential Medium supplemented with 10% foetal bovine serum (GIBCO) and 50 U/ml Pen/Strep (Lonza, Switzerland). C4-I

were grown in Waymouth's MB 752/1 medium supplemented with 10% foetal bovine serum and 50 U/ml Pen/Strep. All other cells were grown in DMEM supplemented with 10% foetal bovine serum and 50 U/mL Pen/Strep. Cells were regularly tested for Mycoplasma and were negative during this investigation. Cell identity was confirmed by STR profiling.

### Plasmids, siRNA and small molecule inhibitors
pMSCV-N-HA-HPV18 E6-IRES-Puro, pMSCV-N-HA-HPV18 E7-IRES-Puro were previously described in refs. 10,40 (plasmid provided by Elizabeth White, University of Pennsylvania, USA). GFP-HPV18 E7 was previously described in ref. 41. pMSCV-N-HA-HPV16 E7-IRES-Puro, pcDNA3.1(+)-N-DYK-SP12TD, pcDNA3.1(+)-N-DYK-Trunc SP1, pcDNA3.1(+)-N-DYK-TAZ, pcDNA3.1(+)-N-DYK-YAP1, pcDNA3.1(+)-N-DYK-TAZ TBM and pcDNA3.1(+)-N-DYK TAZ S89D were generated by Genscript (Cambridge, UK). shRNA targeting TAZ was kindly provided by Henning Wackerhage (Technical University of Munich). *YAP1* and *YAP1/TAZ (WWTR1)* (Y/T) shRNA were cloned as described[42]. *TOGARAM2* targeting shRNA was purchased from Merck. HPV18 E6 and E7 specific siRNA were previously described[41]. Primer sequences are available in Supplementary Data S1. The human *WWTR1* promoter (NM_015472) sequence (1392 bp upstream to 288 bp downstream of transcriptional start site) was cloned into the pEZX-PLO1 backbone upstream of *Firefly* luciferase and with a *Renilla* luciferase tracking gene by GeneCopoeia. The predicted SP1 binding sites (AGGGCGGCAG and ATCCCGCCCC respectively) were deleted using site directed mutagenesis. Primers can be found in Supplementary Data S1. Cyclohexamide (10 μg/mL treatment times indicated), Mithramycin A (50 nM 24 h treatment), 6079510

(10 μM 16 h treatment), Ivermectin (10 μM 24 h treatment) and U0126 (10 μM 24 h treatment) were purchased from Calbiochem, resuspended in DMSO and used as indicated.

## Mammalian cell transfection

Plasmids and siRNA were transfected using Lipofectamine® 2000 (ThermoFisher) at a ratio of 1:2. Cells were processed for western blotting, RNA extraction or reseeded for further analysis as required.

## Generation of stable cell lines

pPAX2, pVSVG and either a pLKO.1-non-targeting shRNA construct or targeting shRNA constructs (either cloned or purchased from Merck, sequences available upon request) were transfected into HEK293TTs in a ratio of 0.65:0.65:1.2 respectively. For overexpression and poly-clonal cell lines, the plasmid of interest was transfected into Phoenix A cells (kindly provided by Garry Nolan, Stanford, USA). 72 h post transfection, complete DMEM media containing lentivirus was removed and filtered with a 0.45 μm regenerated cellulose filter (Sartorius, 16555-K). Appropriate cells were seeded at a density of $7.5 \times 10^5$ in a 10 cm culture dish 24 h prior to use. Media was aspirated and lentivirus containing media was added in a 1:1 ratio with fresh complete DMEM media to cells along with 4 μg/μl Polybrene (Santa Cruz, sc-134220) and 20 mM HEPES. Cells were then incubated for 72 h before the addition of selection antibiotic puromycin. Poly-clonal cell lines were passaged for 1 week before screened by RT-qPCR or western blot for successful transduction. To generate monoclonal stable cell lines, polyclonal stocks were diluted to 1 cell per well manually in a 96 well plate and surviving cells were screened for sufficient knockdown of target gene via RT-qPCR when sufficiently grown.

## Western blot analysis

Equal amounts of protein lysates were separated by SDS-PAGE before transfer onto a nitrocellulose membrane via a semi-dry transfer method (Trans Blot SD Semi-Dry Transfer cell, Bio-Rad, USA). Membranes were then blocked in 5% milk/TBST solution before incubation in primary antibody overnight at 4 °C. Primary antibodies used in this study included: TAZ (BD; 560235, 1:500), YAP (CST; D8H1X, 1:1000), HPV 16 E7 (SCBT; sc-1587, 1:250), HPV 18 E7 (Abcam; ab100953, 1:1000), HPV18 E6 (SCBT G7; sc-365089, 1:500), HPV 16 E6 (GeneTex Inc; GTX132686, 1:500), GFP (SCBT B-2; sc-9996, 1:2500), phospho-ERK1/2 (T202/Y204) (CST; 9101, 1:1000), ERK1/2 (CST; 9102, 1:1000) FLAG (Sigma-Aldrich; F1804, 1:1000) and GAPDH (SCBT; sc365062, 1:5000). Horseradish peroxidase (HRP)-conjugated secondary antibodies (Sigma, USA) were used at a 1:5000 dilution. Proteins were detected using ECL and visualised on X-ray film.

## RNA extraction and RT-qPCR

Total RNA was extracted from cells using the E.Z.N.A.® Total RNA Kit I (Omega Bio-Tek) following the protocol for cultured cells and concentration was determined using NanoDrop™ One spectrophotometer. RT-qPCR was performed using GoTaq Taq 1-Step RT-qPCR System (Promega) and a CFX connect Real-Time System (Biorad) on 10 ng of RNA. The ΔΔCT analysis method was used, normalised to the U6 housekeeper. Primer sequences can be found in Supplementary Data S1.

## Chromatin Immunoprecipitation (ChIP)

Cells were fixed in 1% formaldehyde for 10 min at room temperature, quenched in 0.25 M glycine, and washed in ice-cold PBS. Cells were harvested by scraping and then lysed in cell lysis buffer (10 mM Tris-HCl, pH 8.0, 10 mM NaCl, 0.2% NP-40, 10 mM sodium butyrate, 50 μg/ml phenylmethylsulfonyl fluoride (PMSF), 1× complete protease inhibitor). Nuclei were collected by centrifugation at 2500 rpm at 4 °C and resuspended in nuclear lysis buffer (50 mM Tris-HCl, pH 8.1, 10 mM

EDTA, 1% SDS, 10 mM sodium butyrate, 50 μg/ml PMSF, 1× complete protease inhibitor). Extracted chromatin was then sonicated and chromatin concentration was determined. Approximately 100 μg of chromatin from each sample was used for the experiment. SP1 was immunoprecipitated using a ChIP-grade anti-SP1 antibody (CST; 9389) and each sample was simultaneously pulled down with an IgG isotype control. Protein A/G Sepharose™ (Merck; P3296) was used for antibody-chromatin immunoprecipitations. Chromatin was processed for qPCR using primers probing SP1 binding sites within the *WWTR1* promoter region (Supplementary Data S1). Fold change enrichment was calculated by comparing to the IgG isotype control following normalisation to the input sample[39].

## Proliferation assays

Growth curve analysis, colony formation and soft agar assays were performed as previously described[41,43]. For proliferation assays, cells were detached by trypsinisation and reseeded at equal densities in 12 well dishes. Cells were subsequently harvested every 24 h and manually counted using a haemocytometer. For colony formation assays, cells were detached by trypsinisation after treatment or transfection as required and reseeded at 500 cells/well in six well dishes. Once visible colonies were noted (typically 14 days), culture media was aspirated and cells fixed and stained in crystal violet staining solution (1% crystal violet, 25% methanol) for 15 min at room temperature. Plates were washed thoroughly with water to remove excess crystal violet and colonies counted manually. For soft agar assays, 60 mm cell culture dishes were coated with a layer of complete DMEM containing 0.5% agarose. Simultaneously, cells were detached by trypsinisation after treatment or transfection as required and resuspended at 1000 cells/mL in complete DMEM containing 0.35% agarose and added to the bottom layer of agarose. Once set, plates were covered with culture media and incubated for 14–21 days until visible colonies were observed. Colonies were counted manually. Each experiment was repeated a minimum of three times.

## Wound healing

Cells were seeded at a density of $5 \times 10^5$ and grown until fully confluent. When confluent, cells were wounded with a p200 pipette tip and imaged using an EVOS Auto 2 Microscope (ThermoFisher Scientific). Cells were then incubated for a further 24 h before reimaging. Each condition was imaged 3 times at each time point. Images were analysed in Image J (National Institutes of Health, USA).

## Migration and invasion assay

24 h post transfection cells were trypsinised and reseeded at a density of $2.5 \times 10^4$ in the cell culture insert of a Transwell assay in serum free media. Fresh complete media was added to the well (below the permeable membrane). Cells were then incubated a further 24 h before all media was aspirated and all non-migrated cells (which remain above the membrane) were removed with a cotton swab without puncturing the membrane. Migrated cells were then stained before the membrane were imaged using an EVOS Auto 2 Microscope. For invasion assays, a Transwell insert with matrix was used from the CHEMICON Cell invasion assay kit (Merck ECM550).

## Immunofluorescence analysis

To visualise TAZ, cells were fixed in 4% paraformaldehyde for 10 min at room temperature before permeabilisation with 0.3% Triton/PBS for 10 min at room temperature. Permeabilised cells were then blocked in 10% BSA/0.1%Triton/PBS (10% BSA/PBS-T) at room temperature for 1 h. Cells were immediately inverted and incubated in anti-TAZ primary antibody (1:100) in 2% BSA/PBST overnight in a humid chamber at 4 °C and then incubated with Alexa Fluor 488-conjugated secondary antibody (1:1000) (Invitrogen) in PBS with 10%

BSA for 2 h at room temperature. DAPI was used to visualise nuclei and coverslips were mounted onto slides with Prolong Gold (Invitrogen). All other immunofluorescence analysis was performed as previously described[10].

## Protrusion analysis

Cells were fixed, permeabilised and incubated in primary antibodies overnight at 4 °C[44], then incubated in Rhodamine-Phalloidin (Abcam ab235138) alongside appropriate Alexa Fluor-conjugated secondaries for 2 h at room temperature in 2% BSA/PBS. Cells were next mounted and DAPI was used to visualise nuclei. Protrusion lengths were measured manually using Image J and the number of protrusions per cell was counted. Ten cells were counted over at least 3 repeats for each condition.

## Luciferase reporter assay

Cells were transfected with appropriate *Firefly* luciferase reporter plasmids and control. Samples were lysed at 24 h post transfection with passive lysis buffer (Promega) and activity was subsequently measured in triplicate and normalised to expression of the co-expressed *Renilla* luciferase control using a dual-luciferase reporter system (Promega)[45].

## RNA-sequencing

Library preparation and Human Whole Transcriptome Sequencing was performed by NovaSeq (PE150 & SE5 for mRNA and lncRNA). Reads were quality filtered ($Q < 20$) and adapter trimmed using Trimmomatic (V 0.39)[46]. Processed reads were aligned using human reference genome GRCh38/hg38 assembly using HISAT2 (V 2.1.0)[47]. Expression counts across genomic features were generated using HTSeq (V 0.11.1)[48] on human GRCh38 reference annotation (GENCODE release 32), with parameters -s no, -a 10, -m union and -nonunique none. Raw expression counts were normalised by counts per million (CPM). Differential expression (DE) analysis between two groups were performed using edgeR R package with glmQLFit and glmQLTest functions[49]. The following contrasts were made: NEG vs TAZ-S1 = A1, NEG vs TAZ-S2 = A2, NEG vs YAP-S1 = B1, NEG vs YAP-S2 = B2. The DE mRNAs were defined at adjusted *p*-value < 0.05 using[50] and with a LogFC threshold of 0.5. Due to the differences in the shRNA binding sites, DE analysis was performed on individual shRNA KD samples compared to scrambled shRNA control, each containing two biological replicates. Genes DE in both A1 and A2 comparisons, but not DE in either B1 or B2, were extracted and selected for further analysis. To reduce false discovery rate in the DE analysis we included only transcripts with at least 1 CPM in 2 samples.

Biological processes associated with aberrant TAZ expression were assessed by Gene ontology (GO) analysis using the R package, clusterProfiler[51], and the human Bioconductor annotation database (org.Hs.eg.db). Over-representation test was performed and *p*-value and *q*-value < 0.05 and terms were filtered for redundancies by semantic similarity analysis[52]. Raw data files and processed counts are deposited at GEO (GSE261673).

## Statistics and reproducibility

For data with one independent variable a two-tailed, unpaired Student's *t*-test was used. Correlations were evaluated by Pearson correlation. Differences in Kaplan-Meier survival curves were analysed by the log rank test. The number of replicates and statistical tests are noted for each analysis in the figure legends. Individual *p* values are available in the Source Data file. Unless stated all data presented are representative of at least 3 biological repeats.

## Reporting summary

Further information on research design is available in the Nature Portfolio Reporting Summary linked to this article.

## Data availability

The RNA-sequencing data generated in this study has been deposited in the GEO database under accession code GSE261673. The analysed RNA-sequencing data are available in Supplementary Information. The TCGA data were accessed and are available through cBioPortal. The remaining data are provided with the Article, Supplementary Information or the Source Data file. Source data are provided with this paper.

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

## Acknowledgements

We are grateful to Prof Henning Wackerhage (Technical University of Munich) and Prof Gary Nolan (Stanford, USA) for provision of reagents. We thank the Scottish HPV Investigators Network (SHINE), Prof Sheila Graham (University of Glasgow), Dr David Millan (University of Glasgow) and Prof Nick Coleman (University of Cambridge) for providing HPV+ patient samples. We would like to thank Dr Martin Stacey for provision of the HaCaT cell line. We thank Prof Stephen Griffin for providing verteporfin and critical reading of the manuscript. Work in the Macdonald lab is supported by Medical Research Council (MRC) funding (MR/X009564/1, MR/K012665 and MR/S001697/1). J.A.S. and J.A.C. were funded by Faculty of Biological Sciences, University of Leeds Scholarships. M.R.P. and D.A.T. were funded by Biotechnology and Biological Sciences Research Council (BBSRC) studentships (BB/M011151/1 and BB/T007222/1). R.C. was supported by an MRC DiMeN studentship (MR/W006944/1). M.W. was supported by a University of Leeds China Scholarship. E.L.M. was supported by the Wellcome Trust (1052221/Z/14/Z and 204825/Z/16/Z) and the University of Sussex. The funders had no role in study design, data collection and analysis, decision to publish, or preparation of the paper. Figures 3A and 10 were created with BioRENDER.com.

## Author contributions

Conceptualisation (M.R.P., E.L.M., A.M.); Formal analysis (M.R.P., J.A.S., J.A.C., C.A.A., E.L.M.); Funding acquisition (M.R.P., E.L.M., A.M.); Investigation (M.R.P., J.A.S., J.A.C., E.L.M., R.C., D.A.T., M.W.); Project administration (A.M.); Supervision (A.W., E.L.M. & A.M.); Writing—original draft (M.R.P., E.L.M.); Writing—review & editing (all authors).

## Competing interests

The authors declare no competing interests.
