## [Peer Review File · Nature Communications]

The Hippo pathway transcription factors YAP and TAZ play HPV-type dependent roles in cervical cancerREVIEWER COMMENTS

Reviewer #1 (Remarks to the Author):

In this manuscript, authors demonstrate that TAZ expression was only significantly increased in HPV18 positive cervical cancer cells. knockdown of TAZ expression impeded proliferation, migration and invasion only in HPV18 positive cells. RNA-sequencing of HPV18 positive cervical cancer cells revealed that YAP and TAZ have distinct target profiles, suggesting they promote oncogenic phenotypes via different mechanisms. Furthermore, authors identified TOGARAM2 as a previously uncharacterized TAZ-dependent gene and demonstrate its key roles in migration and invasion in HPV18 positive cancers.

1. Figure 2, how was the TAZ promoter luciferase reporter built?
2. Figure 3M, does YAP expression level changed in this condition?
3. Figure 4A & B, two compounds treatment reduced TAZ/YAP expression rather than led to TAZ exclusion from nuclear; Figure 4B, 6079510 has no obvious effects on YAP.
4. Figure 4E & H, how long does it take to grow colonies in soft agar?
5. Figure 6A, the exogenous YAP expression was much lower than TAZ, thus may explain why YAP expression could not rescue TAZ-KD phenotype.
6. Figure 7, the description for RNAseq analysis was not clear, the differential gene expression fold changes and statistical analysis need to be included.
7. Figure 7H, the clinical relevance of TOGARAM2 expression was not significant.
8. The oncogenic contribution of TAZ to HPV 16 and HPV 18 cervical cancer tumor formation need to be determined in vivo.

Reviewer #2 (Remarks to the Author):

This report demonstrates that HPV18 positive cervical cancer cells are preferentially dependent upon TAZ expression when compared with HPV16 counterparts. This is significant as HPV18+ cc is particularly aggressive with worse clinical outcomes when compared with that of HPV16 (and other high risk HPV). The work is very thorough and a variety of approaches have been taken to demonstrate the main point of the manuscript. The results are all statistically analyzed and the methodology (for the most part) is sound. I have a few specific comments the authors should address.

1. In Figures 1A-C are the cancers with high TAZ in TCGA predominantly HPV18+? This is an important point as splitting the samples into HPV18+ and HPV18- can allow for a false equivalence (i.e. high and low levels in 18+ may be different from high and low in 18-).
2. In Figure 1A please list the cancers on the X axis in the legend.
3. In Figure 1E change the colors, they are too similar and it makes the reader work more than they have to. i.e. change the - to a color and shade them, same with 16 and 18.
4. Figure 2C. I couldn't find a description of the WWTR1 reporter plasmid anywhere, apologies if I overlooked it. If it isn't there then the authors should describe the coordinates of the promoter and how it was cloned into the luciferase reporter.
5. In the C33a cells, does HPV16 E7 increase WWTR1 expression? If it doesn't it emphasizes the importance of 18 E7 for the upregulation in the cancer cells. When studied in isolation the oncogenes can sometimes give misleading results and confirming that HPV16 E7 does not do this would help convince the reader that it is 18 E7 that is upregulating in the cancer cells.
6. In 3D&E, why did you change to GFP-E7 over HA-E7 in figure 2? Are these experiments transient transfection or were stable cell lines used?
7. In lines 370-371 would it be better to state that E7 enhancement of WWTR1 expression is dependent on the ERK1/SP1 pathway? It doesn't seem that E7 is upregulating ERK P so this probably isn't what mediates it exclusively.
8. In Figure 4A and B it seems like the drugs degrade the YAP and TAZ proteins when they are removed from the nucleus. Were western blots carried out to determine if this is the case?
9. In Figure 4 when describing the colony formation assays put agar in the Y-axis description of the data when agar colony formation is described. It helps the reader differentiate between the two assays.

10. In lines 393-397 the authors discuss E7 promoting growth of HaCaT and C33a cells. But they haven't shown growth, only colony formation. I'm not sure these figures add much but if they are staying in then change the discussion a little to reflect that it is colony formation that is being studied, not growth.

11. What happens to HPV16 cells if you knock down TOGARAM2? Would also be interesting to look at other 18 lines and not just HeLa cells.

Reviewer #3 (Remarks to the Author):

The manuscript by Patterson et al provides convincing evidence that TAZ plays an important role in HPV 18 carcinogenesis by regulating proliferation and invasion by HPV 18, but not in HPV 16 carcinogenesis. In contrast YAP, the better studied paralog of TAZ is upregulated in all HPV+ cancers. They confirm data by others that HPV 18 E7 activates Mek signaling and the transcription factor SP1 to regulate TAZ. Mutation of SP1 binding sites of a dominant negative SP1 inhibits TAZ transcription. Using RNAseq they show that YAP and TAZ have separate, though partially overlapping transcriptomic signatures. They focus on one gene regulated by TAZ, TOGARAM2 and show that it is at least partially responsible. Although the effects are small, they are convincing. However, what is lacking is any mechanistic explanation of how 18E7 but not 16E7 regulates TAZ, as 16 E7 also activates MEK/ERK signaling and induces SP1.

Main points:

1. A more thorough examination of the TAZ promoter is needed in 18 E7 vs. 16 E7 cells is needed. Is SP1 bound to both equally? Are the histone modifications the same? What other proteins are at the promoter?

2. Does 18 E7 activate TAZ transcription in primary cells or only in the context of transformed cells?

3. Since KD of E6/E7 stops proliferation wouldn't it KD YAP also (Fig. 2A).

3. The color scheme of black, dark green, blue for the figures is very difficult to see. For example, 1E, 5A, 5F. Similarly, the microscopy is hard to see.

Minor points:

1. What are the tumor types in Fig 1A?

2. Figure Legend for 1 G+H – line 778. What does three biological repeats mean in this context?

3. What are the sequences for the proposed SP1 binding sites? What is the consensus site? What mutations were made?

REVIEWER COMMENTS

We would like to thank the reviewers for their positive and constructive comments. We have provided a point-by-point response below. All figure numbers correspond to the updated manuscript, which has text changes highlighted.

Reviewer #1 (Remarks to the Author):

In this manuscript, authors demonstrate that TAZ expression was only significantly increased in HPV18 positive cervical cancer cells. knockdown of TAZ expression impeded proliferation, migration and invasion only in HPV18 positive cells. RNA-sequencing of HPV18 positive cervical cancer cells revealed that YAP and TAZ have distinct target profiles, suggesting they promote oncogenic phenotypes via different mechanisms. Furthermore, authors identified TOGARAM2 as a previously uncharacterized TAZ-dependent gene and demonstrate its key roles in migration and invasion in HPV18 positive cancers.

1. Figure 2, how was the TAZ promoter luciferase reporter built? *This plasmid was cloned by GeneCopoeia. Sequences from 1392 bp upstream to 288bp downstream of the WWTR1 transcriptional start site were cloned into a luciferase reporter plasmid. We have updated the methods section to include this information.*

2. Figure 3M, does YAP expression level changed in this condition? *We saw no change in YAP protein levels in the multiple repeats of this experiment in which we blockade ERK1/2 activation using UO126. This is in contrast to the significant loss of TAZ expression under the same conditions. We have now included a YAP blot in Figure 3M and refer to the additional YAP data in the results (line 451).*

3. Figure 4A & B, two compounds treatment reduced TAZ/YAP expression rather than led to TAZ exclusion from nuclear; Figure 4B, 6079510 has no obvious effects on YAP. *We used these two inhibitory compounds because of their published effects on reducing TAZ and YAP nuclear translocation. 6079510 has been shown to target TAZ whereas Ivermectin has broad YAP/TAZ effects (Nagashima et al., 2021; Nisho et al., 2016). Our immunofluorescence data in Fig 4A/4B confirms this, with only Ivermectin reducing YAP nuclear localisation. Based on this reviewer comment, we did run a western blot to test the impact on TAZ/YAP protein levels (see below). We saw a small decrease in TAZ protein expression with both inhibitors while neither inhibitor reduced YAP expression. It is not surprising that TAZ is partially degraded as this commonly occurs with cytoplasmic TAZ. We felt that given the clear loss of nuclear localisation that the immunofluorescence data alone was sufficient to show for the inhibitors. However, if the reviewer feels that the work would be strengthened by also including the additional western blot (provided for the rebuttal) then we are happy to include this as part of the revision process.*

Rebuttal Fig 1. Western blot of TAZ and YAP protein levels in HeLa cells treated with nuclear translocation inhibitors.

4. Figure 4E & H, how long does it take to grow colonies in soft agar? *It takes 2-3 weeks depending on the cell line in question. We have now included this information in the methods section (lines 272-273).*

5. Figure 6A, the exogenous YAP expression was much lower than TAZ, thus may explain why YAP expression could not rescue TAZ-KD phenotype. *We apologise for the confusion here. Figure 6A shows that the expression of exogenous FLAG tagged YAP is higher than the expression of any of the exogenous FLAG TAZ proteins and can be seen on a shorter FLAG exposure (labelled FLAG (short) in figure). If the reviewer feels that our labelling or text remains unclear, we are happy to modify if necessary.*

6. Figure 7, the description for RNAseq analysis was not clear, the differential gene expression fold changes and statistical analysis need to be included. *Fold changes and statistical analysis values associated with TOGARAM2, as well as additional clarity when referring to specific comparisons, have been added to the main text (e.g. line 557, line 556-569). The contrasts made by differential expression analysis, as well as the thresholds defining significance, are detailed in the RNA-Sequencing methods section.*

7. Figure 7H, the clinical relevance of TOGARAM2 expression was not significant. *We agree that the clinical relevance of TOGARAM2 is not statistically significant in terms of progression free survival; however, as we found it does trend in a similar manner to TAZ, we found it interesting and worth including in the manuscript. We have taken care to not call these results significant. We are encouraged by the observation that TOGARAM2 mRNA was significantly increased in high grade HPV18+ CIN patient samples (Fig 7F) and that this strongly correlates with the TAZ expression profile (Fig 7G).*

8. The oncogenic contribution of TAZ to HPV 16 and HPV 18 cervical cancer tumor formation need to be determined in vivo. *We agree with the reviewer that this would be an extremely interesting experiment to perform. Sadly, we do not have the funding to do this experiment right now and we feel it would be better placed in follow up work where we can properly address the contribution of YAP and TAZ across the different HPV types to tumour formation in vivo.*

Reviewer #2 (Remarks to the Author):

This report demonstrates that HPV18 positive cervical cancer cells are preferentially dependent upon TAZ expression when compared with HPV16 counterparts. This is significant as HPV18+ cc is particularly aggressive with worse clinical outcomes when compared with that of HPV16 (and other high risk HPV). The work is very thorough and a variety of approaches have been taken to demonstrate the main point of the manuscript. The results are all statistically analyzed and the methodology (for the most part) is sound. I have a few specific comments the authors should address.

1. In Figures 1A-C are the cancers with high TAZ in TCGA predominantly HPV18+? This is an important point as splitting the samples into HPV18+ and HPV18- can allow for a false equivalence (i.e. high and low levels in 18+ may be different from high and low in 18-). *We performed this analysis but found no significant difference in the dataset used, however, the dataset is skewed and includes largely HPV16+ samples. Additionally, in Figure 1A, TAZ was found to be amplified in many cancers not linked to HPV infection.*

2. In Figure 1A please list the cancers on the X axis in the legend. *This has now been included in the figure legend due to limited space in the figure itself.*

3. In Figure 1E change the colors, they are too similar and it makes the reader work more than they have to. i.e. change the - to a color and shade them, same with 16 and 18. *We have changed the colours while keeping it in line with the rest of the paper. Hopefully this is now easier for the reader.*

4. Figure 2C. I couldn't find a description of the WWTR1 reporter plasmid anywhere, apologies if I overlooked it. If it isn't there then the authors should describe the coordinates of the promoter and how it was cloned into the luciferase reporter. *Our apologies, this was an omission. This plasmid was cloned by GeneCopoeia. Sequences from 1392 bp upstream to 288bp downstream of the WWTR1 transcriptional start site were cloned into a luciferase reporter plasmid. We have updated the methods section to include this information.*

5. In the C33a cells, does HPV16 E7 increase WWTR1 expression? If it doesn't it emphasizes the importance of 18 E7 for the upregulation in the cancer cells. When studied in isolation the oncogenes can sometimes give misleading results and confirming that HPV16 E7 does not do this would help convince the reader that it is 18 E7 that is upregulating in the cancer cells. *This experiment was in our initial submission (Fig 2G-I) and we agree that it is very exciting to see a difference between HPV16 E7 and HPV18 E7 proteins. We also saw no effect on TAZ mRNA or protein levels when HPV16 E6/E7 were depleted by siRNA from the CaSKi cell line (Fig S2 D-E). These data do strongly reinforce different functions for the E7 protein in this instance.*

6. In 3D&E, why did you change to GFP-E7 over HA-E7 in figure 2? Are these experiments transient transfection or were stable cell lines used? *Initially we did use the HA-E7 in these experiments; however, we found that Mith A treatment had the unexpected effect of reducing expression of the HA-E7 protein from our stable cells, making it difficult to discern the effects of SP1 from general knockdown of E7 (rebuttal Fig 2). We could not explain this finding and troubleshooting did not resolve it, and so tested different E7 expression systems available in the lab. The transient expression of GFP-E7 was not affected by Mith A and so we used that approach for this part of the work. We have now noted that it was transient expression in the manuscript text (line 435). Such unexpected effects are why we use additional approaches including mutants and siRNA/shRNA knockdowns to solidify our findings.*

Rebuttal Fig 2. Effect of Mith A on HA-E7 expression.

7. In lines 370-371 would it be better to state that E7 enhancement of WWTR1 expression is dependent on the ERK1/SP1 pathway? It doesn't seem that E7 is upregulating ERK P so this probably isn't what mediates it exclusively. *We took on board your comment and resolved to*

repeat our experiments as we had often seen increases in ERK phosphorylation in E7 expressing cells. In multiple repeats we now have data showing a clearer increase in P-ERK in the E7 expressing cells that is very clearly reduced in cells treated with UO126 (Fig 3M). More broadly, we take on your comment that SP1 can be regulated by kinases other than ERK and so we cannot exclusively state that this is the only mechanism. As such we have tempered our language to make this point clearer (line 678). We feel that future studies should really focus on the manner of TAZ regulation by E7 since that is not the major focus of this manuscript – which aims to highlight the novel differences between TAZ/YAP between different HPV driven cancers and the potential roles of their target genes such as TOGARAM2.

8. In Figure 4A and B it seems like the drugs degrade the YAP and TAZ proteins when they are removed from the nucleus. Were western blots carried out to determine if this is the case? *Please see response to reviewer 1 comment 3 and Rebuttal Fig 1.*

9. In Figure 4 when describing the colony formation assays put agar in the Y-axis description of the data when agar colony formation is described. It helps the reader differentiate between the two assays. *We have made changes to the figures where necessary.*

10. In lines 393-397 the authors discuss E7 promoting growth of HaCaT and C33a cells. But they haven't shown growth, only colony formation. I'm not sure these figures add much but if they are staying in then change the discussion a little to reflect that it is colony formation that is being studied, not growth. *We have altered the text to better reflect our results.*

11. What happens to HPV16 cells if you knock down TOGARAM2? Would also be interesting to look at other 18 lines and not just HeLa cells. *When we performed growth curve analysis of TOGARAM2 knockdown HPV16+ SiHa cells we found no significant effect on proliferation. This data has been included in the manuscript (Fig S10). Also, we have now backed up our HeLa data by generating TOGARAM2 knockdown in the HPV18+ SW756 cell line; crucially, we find a very similar impact on proliferation. This has now been included in the manuscript also in Fig S10.*

Reviewer #3 (Remarks to the Author):

The manuscript by Patterson et al provides convincing evidence that TAZ plays an important role in HPV 18 carcinogenesis by regulating proliferation and invasion by HPV 18, but not in HPV 16 carcinogenesis. In contrast YAP, the better studied paralog of TAZ is upregulated in all HPV+ cancers. They confirm data by others that HPV 18 E7 activates Mek signaling and the transcription factor SP1 to regulate TAZ. Mutation of SP1 binding sites of a dominant negative SP1 inhibits TAZ transcription. Using RNAseq they show that YAP and TAZ have separate, though partially overlapping transcriptomic signatures. They focus on one gene regulated by TAZ, TOGARAM2 and show that it is at least partially responsible. Although the effects are small, they are convincing. However, what is lacking is any mechanistic explanation of how 18E7 but not 16E7 regulates TAZ, as 16 E7 also activates MEK/ERK signaling and induces SP1.

Main points: 1. A more thorough examination of the TAZ promoter is needed in 18 E7 vs. 16 E7 cells is needed. Is SP1 bound to both equally? Are the histone modifications the same? What other proteins are at the promoter? *Previously we had shown a differential effect of HPV16 E7 and HPV18 E7 on TAZ levels. Our investigation showed that both a small molecule inhibitor of SP1 and a dominant negative version of the protein reduced TAZ levels in a HPV18+ cell line and in HPV- cells expressing HPV18 E7. Furthermore, we could show that expression of a constitutive active SP1 could rescue the loss of TAZ expression seen in*

HPV18 E6/E7 knockdown cells. Mechanistically, a luciferase reporter plasmid containing deletions in the predicted SP1 binding sites could no longer be activated by E7. All of these data strongly pointed to SP1 playing a key role in the increased TAZ expression by E7. In revisions, we have performed ChIP-qPCR analysis and found a significantly greater enrichment of SP1 at the predicted SP1 sites in the TAZ promoter (there is more enrichment at the second site) in HPV18+ compared to HPV16+ cells (Fig 3N). Together with our existing data we feel that this provides a strong case highlighting this pathway as important.

We agree that longer term a more comprehensive analysis both of the pathways leading to SP1 activation and how we see HPV18 specific upregulation of SP1 at the TAZ promoter is worth undertaking. However, this would be a significant body of data to fully characterise these processes and we feel that it risks diluting the key take home points of this manuscript that increased TAZ expression is important for HPV18+ cervical cancers, that TAZ and YAP display differential functions between HPV16 and HPV18 positive cancers and that we can learn new insights into the targets of TAZ and YAP such as TOGARAM2.

2. Does 18 E7 activate TAZ transcription in primary cells or only in the context of transformed cells? *This is a very interesting question, as recent beautiful data from the Elizabeth White lab has shown a role for YAP during the productive life cycle. From our data, we show that in CIN1 low grade patient lesions – where we would expect to still see productive replication – there is no significant increase in TAZ levels (Fig 1F). We only really see a significant increase in the high-grade lesions, indicative of TAZ being (more) important in disease rather than replication. In unpublished data, we have looked at TAZ/YAP and classical Hippo-regulated mRNA expression in primary keratinocytes harbouring HPV18 (Rebuttal Fig 3). Whilst we see increases in AREG and CYR61 levels (classic Hippo-dependent genes) we did not observe an increase in TAZ levels. Given that our studies suggest that TAZ is transcriptionally regulated, we take these data to mean that TAZ may not be up-regulated in primary culture models or in patient samples indicative of a productive infection. This suggests TAZ may principally play a role in disease.*

Rebuttal Figure 3. RT-Quantitative PCR analysis of TAZ/YAP and Hippo-dependent genes in normal primary human keratinocytes and HPV18-containing keratinocytes.

3. Since KD of E6/E7 stops proliferation wouldn't it KD YAP also (Fig. 2A). *We have performed this experiment and found that in HPV18+ HeLa cells siRNA depletion of E6/E7 resulted in a significant reduction in TAZ expression (as we expected) but did not significantly reduce YAP levels. This was an unexpected result given that others have predicted a role for E6/E7 in stabilising YAP. Western blot confirmed effective knockdown of E6/E7 (Fig 2A). We repeated this experiment in a HPV16+ cell line to rule out cell line specific effects. Whilst we saw no effect on the low levels of TAZ in these cells (as we expected) we also saw no significant*

reduction in YAP levels, again our E6/E7 western blot confirmed successful knockdown (Fig S2 D-E). At this point we have no clear explanation for this result, it may be that there are differences between the roles of the oncoproteins in the context of a primary lifecycle compared to changes in the host cell during transformation. We have now added elements of this into the discussion (lines 688). These findings do suggest that when we knockdown E6/E7, our results in the HPV18+ cells are likely caused by loss of TAZ and not YAP.

3. The color scheme of black, dark green, blue for the figures is very difficult to see. For example, 1E, 5A, 5F. Similarly, the microscopy is hard to see. *We have hopefully now amended the colour scheme and microscopy issues.*

Minor points: 1. What are the tumor types in Fig 1A? *This has now been outlined in the figure legend.*

2. Figure Legend for 1 G+H – line 778. What does three biological repeats mean in this context? *Thank-you for highlighting this. It was an error and we have now fixed it.*

3. That are the sequences for the proposed SP1 binding sites? What is the consensus site? What mutations were made? *We have added details to the methods section. The sequences of the predicted SP1 binding sites were AGGGCGGCAG and ATCCCGCCCC respectively located at the sites indicated in Figure 3A. The consensus sequence of SP1 is 5'-(G/T)GGGCGG(G/A)(G/A)(C/T)-3' (Nagaoka et al. 2001). The predicted binding sites were deleted.*

References

Nagashima, S. et al. CSE1L promotes nuclear accumulation of transcriptional coactivator TAZ and enhances invasiveness of human cancer cells. *J Biol Chem* **297(1)**, 100803 (2021)

Nishio, M. et al. Dysregulated YAP1/TAZ and TGF β signalling mediate hepatocarcinogenesis in Mob1a/1b deficient mice. *PNAS* **113(1)**, E71-80 (2016)

Nagaoka et al. (2001). Selected base sequence outside the target binding site of zinc finger protein Sp1. *Nucleic Acid Res.* 29(24):4920-4929.

REVIEWERS' COMMENTS

Reviewer #2 (Remarks to the Author):

This is a resubmission of a previous manuscript investigating the role of TAZ specifically in HPV18 E7 transformation, and not HPV16. This is an extremely interesting and important observation. The authors have rigorously addressed all of the critiques including carrying out a significant number of new experiments. This has strengthened what was already an excellent set of work and I thank the authors for addressing all of the points raised.

Reviewer #3 (Remarks to the Author):

The authors have responded thoughtfully and thoroughly to all my concerns.